



# Polarimetric radar reveals the spatial distribution of ice fabric at domes in East Antarctica

M. Reza Ershadi[1], Reinhard Drews[1], Carlos Martín[2], Olaf Eisen[3,5], Catherine Ritz[4], Hugh Corr[2], Julia Christmann[3,6], Ole Zeising[3,5], Angelika Humbert[3,5], and Robert Mulvaney[2]

[1]Department of Geosciences, University of Tübingen, Tübingen, Germany
[2]British Antarctic Survey, Natural Environment Research Council, Cambridge, UK
[3]Alfred Wegener Institute Helmholtz-Centre for Polar- and Marine Research, Bremerhaven, Germany
[4]University Grenoble Alpes, CNRS, IRD, IGE, Grenoble, France
[5]Department of Geosciences, University of Bremen, Bremen, Germany
[6]Institute of Applied Mechanics, University of Kaiserslautern, Germany

**Correspondence:** Mohammadreza Ershadi (mohammadreza.ershadi@uni-tuebingen.de)

**Abstract.** Ice crystals are mechanically and dielectrically anisotropic. They progressively align under cumulative deformation, forming an ice crystal orientation fabric that, in turn, impacts ice deformation. However, almost all the observations of fabric are from ice core analysis and its interplay with the flow is unclear. Here, we present a non-linear inverse approach that combines radar polarimetry with vertical changes in anisotropic reflection to extract, for the first time, the full orientation tensor. The

orientation tensor is routinely used to synthesize fabric information and it is used in anisotropic ice flow models. We validate our approach at two Antarctic ice-core sites (EDC and EDML) in contrasting flow regimes. Spatial variability of ice-fabric characteristics in the dome-to-flank transition near Dome C is quantified with 20 more sites located along a 36 km long cross-section. Local horizontal anisotropy increases under the dome summit and decreases away from the dome summit. We suggest that this is a consequence of the non-linear rheology of ice also known as Raymond effect. On larger spatial scales, horizontal

anisotropy increases with increasing distance from the dome. At most of the sites, the main driver of ice-fabric evolution is vertical compression, yet our data show that ice fabric horizontal distribution is consistent with the present horizontal flow. Our method, which uses co- and cross polarimetric radar data suitable for profiling radar applications, can constrain ice-fabric distribution on a spatial scale comparable to ice flow observations and models.

## 1 Introduction

The movement of glaciers and ice sheets has two components: ice deformation and basal sliding. Satellites provide widespread and increasingly well-resolved temporal surface velocities. In most cases, however, it is difficult to differentiate the contribution of ice deformation and basal sliding. This results in increased uncertainty in several areas, such as ice-flow model initialization with data assimilation techniques (Schannwell et al., 2019) or predicting erosion rates from surface velocities (Headley et al.,





2012; Cook et al., 2020). Even in ice-sheet covered areas where basal sliding can certainly be excluded, e.g., near ice domes or beneath ice rises (Matsuoka et al., 2015), knowledge of internal ice deformation is important for predicting age-depth relationships for new ice-core drill sites (Parrenin et al., 2007; Martín et al., 2009; Martín and Gudmundsson, 2012) or for using internal layer architecture to reconstruct paleo-ice dynamics (Matsuoka et al., 2015). The temperature-dependent, non-linear, and anisotropic rheology of ice governs how ice deforms and poses many challenges to numerical ice-flow models.

Most models do not consider ice-fabric anisotropy because this quantity is yet poorly constrained by observations. The most reliable observations of ice fabric come from the analysis of ice core thin sections using fabric analyzers detecting single ice crystals' lattice orientation using transmitted light microscopy (Durand et al., 2009; Weikusat et al., 2017). The underlying principle used is that single ice crystals are uniaxially birefringent for electromagnetic waves. This causes the polarization-dependent formation of ordinary and extraordinary waves that propagate through the lattice and superimpose with a phase

shift at the detector. Constructive and destructive superposition of these waves can be used to characterize ice fabric in thin sections at a vertical spacing of centimeters to decimeters (Kerch et al., 2020). Ice penetrating radar on ice sheets employs the same principles, although spatial scales and applied electromagnetic frequencies are different. As will be explained in more detail (Sect. 3.3), ground-penetrating radar systems such as the ground-based Autonomous phase-sensitive Radio Echo Sounder (ApRES) (Brennan et al., 2014; Nicholls et al., 2015) can detect the polarization-dependent phase shift induced by

ice birefringence and also quantify the degree of anisotropic scattering which may be caused by abrupt vertical changes in ice fabric. Other geophysical methods to detect ice-fabric anisotropy are sonic logging of boreholes (Gusmeroli et al., 2012; Pettit et al., 2007) or surface-based seismic surveys (Diez and Eisen, 2015; Diez et al., 2015; Smith et al., 2017; Brisbourne et al., 2019).

Ice core and borehole based methods are reliable and can be obtained in a high vertical resolution. However, in deep ice

where grains may be large compared with the typical ice-core diameter of 10 cm, they are statistically not well constrained. They also do not provide much spatial context and are often obtained at dome locations where the horizontal advection is negligible and the climate record is easier to interpret. The majority of radar profiles are not analyzed with respect to ice fabric anisotropy often because the radar systems do not provide the required precision or are collected with a single polarization only. The collection of crossing radar lines partially remedies this issue. However, newer radar systems collect data with cross-

polarized arrays so that area-wide detection of ice anisotropy appears to be a target within reach (Yan et al., 2020). The theory of radar birefringence in glaciology has long been known (Hargreaves, 1978; Woodruff and Doake, 1979; Matsuoka et al., 1997; Fujita et al., 1999), and has recently been significantly extended to exploit the capacity of phase information from newer radar systems that were previously not available (Dall, 2010; Jordan et al., 2019, 2020). Examples for applications of radar polarimetry exist near ice domes in Greenland (Gillet-Chaulet et al., 2011; Jordan et al., 2019) and Antarctica (Fujita et al.,

1999; Brisbourne et al., 2019), on ice rises (Drews et al., 2015; Matsuoka et al., 2015; Brisbourne et al., 2019), in flank-flow regimes (Eisen et al., 2007), divides (Young et al., 2020), and for ice streams (Robert et al., 1993; Joughin et al., 1999; Jordan et al., 2020). However, there is not yet a clear observation-based picture of how ice fabric develops across the different flow regimes.



Here, we built on a previously derived forward modeling framework (Fujita et al., 2006), which was extended by Jordan

et al. (2019, 2020). We extend it further with theory relating to anisotropic reflections and then develop an inverse approach that also attempts to characterize ice-fabric types continuously along depth and for all of the three bulk crystallographic axes. We demonstrate this for 20 ApRES sites covering the dome-flank transition near the EPICA-Dome C (EDC) ice core and an additional location at the EPICA-DML (EDML) ice-core site in eastern Dronning Maud Land.



**Table 1.** Important variables sorted in order of appearance.

| Symbol | Unit | Description |
|---|---|---|
| $\mathbf{v}$ | - | Ice fabric Eigenvector |
| $\lambda$ | - | Ice fabric Eigenvalue |
| $\varepsilon'$ | - | Dielectric permittivity matrix |
| $\mathbf{e}$ | - | Electric field vector |
| H, V | - | Horizontal and Vertical polarization plane |
| TR | - | Tx-Rx aerial line |
| $\theta$ | ° | Ice fabric orientation |
| $\alpha$ | ° | Georeferencing angle |
| $z$ | m | Depth (0 at the surface, positive downward) |
| $i$ | - | Stratified ice layer index |
| $N$ | - | Number of layers |
| $\mathbf{T}$ | - | Transmission matrix |
| $k_x, k_y$ | - | Wavenumbers along the two principal axes |
| $\mathbf{\Gamma}$ | - | Reflection matrix |
| $\mathbf{S}$ | - | Scattering matrix |
| $s_{HH}, s_{VV}$ | - | Complex co-polarized scattering signals |
| $s_{HV}, s_{VH}$ | - | Complex cross-polarized scattering signals |
| $\mathbf{R}$ | - | Rotation matrix |
| r | - | Reflection ratio |
| $\Delta\lambda$ | - | Ice fabric horizontal anisotropy |
| $\phi$ | rad | Phase difference |
| $C_{HHVV}$ | - | Complex polarimetric coherence |
| $\phi_{HHVV}$ | rad | Polarimetric coherence phase |
| $\Psi$ | - | Scaled phase derivative |
| $P$ | dB | Power anomaly |
| $n$ | - | Number of angular increments |
| $AD$ | ° | Nodes angular distance |
| $J$ | - | Cost function |





## 2 Study areas

We use radar data near two deep ice-core drill sites in East Antarctica. One is located at Dronning Maud Land (DML), near the German summer station (Kohnen at $0.00°, -75.00°$ S). The other site is located at Dome C, close to Concordia station ($123.35°$ E, -75.10° S). We use the measured ice-fabric data from both ice cores published by Weikusat et al. (2017) and Durand et al. (2009), respectively, to validate our polarimetric-radar data inferences. At Dome C, data were additionally collected at 20 stations along with a 36 km long profile across the dome, enabling us to track ice-fabric variability in the dome-flank

transition zone. Surface topography at Dome C (Helm et al., 2014; Howat et al., 2019) exhibits an ice dome elongated in the SW-NE direction (Fig. 1a). Surface velocities are too slow ($<0.02$ ma$^{-1}$) for reliable detection with satellite imagery. GPS measurements show that ice-flow direction follows the surface maximum gradient direction, increases with distance from the dome, and is near-parallel to the transect described above (Vittuari et al., 2004). Kohnen station is located near a transient ice-divide triple junction in a flank-flow regime, and the ice flow is significantly faster ($\approx 0.74$ ma$^{-1}$) than at Dome C. At

Dome C, the largest principal strain rate is oriented SW-NE (Rémy and Tabacco, 2000; Vittuari et al., 2004). At EDML, the maximum strain rate is oriented along $24°$ N (Wesche et al., 2007; Drews et al., 2012).





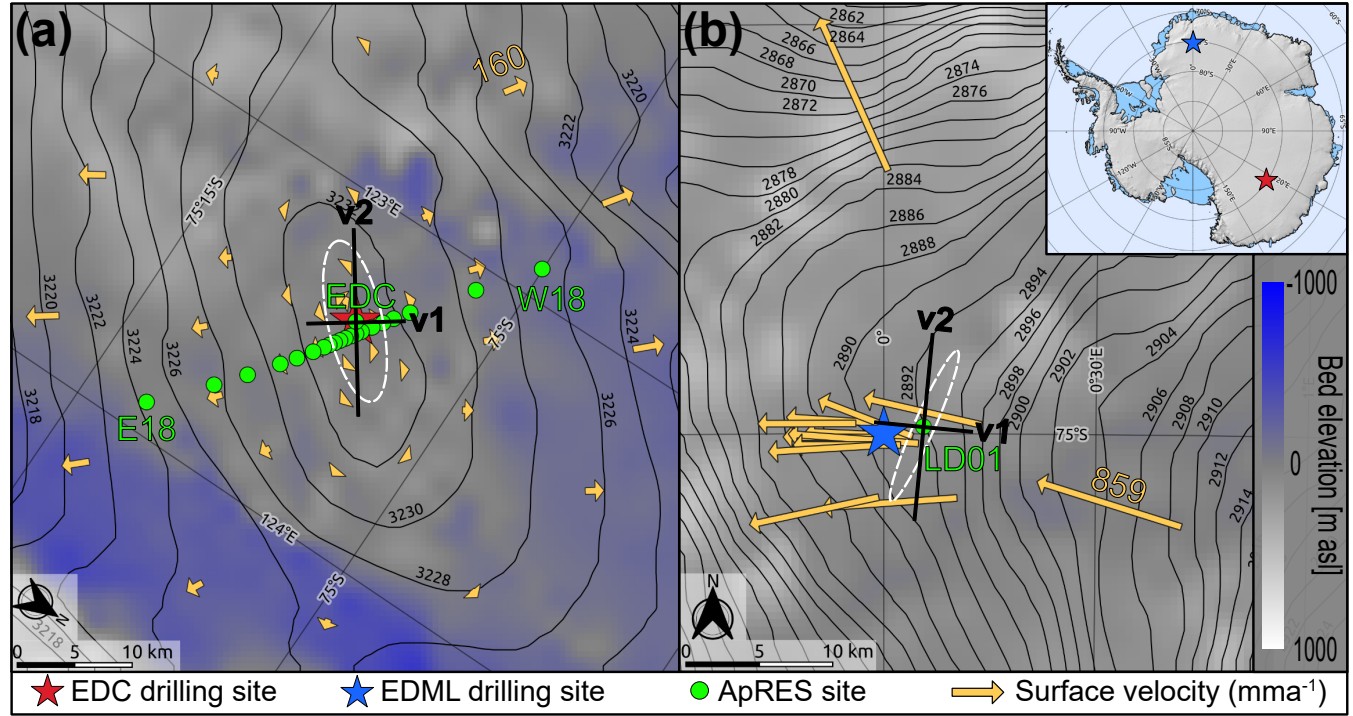

**Figure 1.** Map of the study areas. (a) EPICA Dome C (EDC). (b) EPICA Dronning Maud Land (EDML). Black contour lines are the surface elevation (Helm et al., 2014). The background color is the bed elevation (Morlighem et al., 2020). Yellow arrows are the magnitude and direction of the surface velocities for Dome C (Vittuari et al., 2004) and EDML (Wesche et al., 2007). The white strain ellipses mark the directions of the maximum and minimum strain rate. **v1** and **v2** are the ice-fabric's horizontal Eigenvectors, and they are based on the results in Sects. 4.1 and 4.2. Note that (a) and (b) have a different scale and orientation.

## 3 Methods

### 3.1 Quantitative metrics used to define the ice fabric

Ice crystallizes in the shape of hexagons and the direction normal to the basal plane is described with the c-axis (Hooke, 2005).
In a given strain regime, individual ice crystals orient themselves so that the bulk c-axis orientation of many crystals forms a distinct pattern which we refer to as ice fabric. Elsewhere it is also described with Crystal Orientation Fabric (COF) or Lattice Preferred Orientation (LPO) (Weikusat et al., 2017). Development of ice fabric can lead to ice softening along with certain directions by up to a factor of 60 (Duval et al., 1983). The mechanical anisotropy is accompanied by a dielectric anisotropy to which the radio waves are sensitive described by the orientation tensor (Gödert, 2003; Gillet-Chaulet et al., 2006; Martín et al.,
2009). The bulk ice-fabric pattern is often quantified using the Eigenvectors ($\mathbf{v1}, \mathbf{v2}, \mathbf{v3}$) and Eigenvalues ($\lambda1, \lambda2, \lambda3$) of an ellipsoid that best represents the c-axis orientation of all ice crystals in the sample. The Eigenvalues are normalized

$$\lambda1 + \lambda2 + \lambda3 = 1, \tag{1}$$





and to be consistent with the past polarimetric radar studies, we assume

$$\lambda 1 < \lambda 2 < \lambda 3. \tag{2}$$

Combination of Eqs. (1) and (2) set bounds on the Eigenvalues ($0 \leq \lambda 1 \leq 0.33$, $0 \leq \lambda 2 \leq 0.5$, and $0.33 \leq \lambda 3 \leq 1$). The Eigenvalues can be used to distinguish the ice-fabric types such as isotropic ($\lambda 1 \approx \lambda 2 \approx \lambda 3$), girdle ($\lambda 1 \ll \lambda 2 \approx \lambda 3$), and single maximum ($\lambda 1 \approx \lambda 2 \ll \lambda 3$) (Woodcock, 1977; Azuma, 1994; Fujita et al., 2006). The Eigenvalues and Eigenvectors can be used to describe the dielectric permittivity tensor $\varepsilon'$, containing the bulk permittivities $\epsilon'_x, \epsilon'_y, \epsilon'_z$ relevant for radio-wave propagation (Sect. 3.3).

## 3.2  Data collection

The radar data in this study were collected using a phase-sensitive frequency-modulated continuous-wave radar system (Brennan et al., 2014; Nicholls et al., 2015) with a 200 MHz bandwidth and $f_c = 300$ MHz center frequency. This radar emits linearly polarized electromagnetic waves using a slot antenna where the direction of the polarization plane is aligned with the direction of the electric field vector (e) in the antenna as shown in Fig. 2a.

We use terminology from satellite radar polarimetry to distinguish the directions of the polarization with H and V, although, in a nadir-looking geometry, these are arbitrarily determined because H and V both have horizontal polarization plane at depth. Here, we name the horizontal (H) and vertical (V) polarization plane consistent with Jordan et al. (2019). However, we want to point out that this definition is different to the one applicable to seismic shear waves, where vertically receiver (thus having a vertical component upon reflection at depth for non-vertical angles of incidence), and vice-versa.



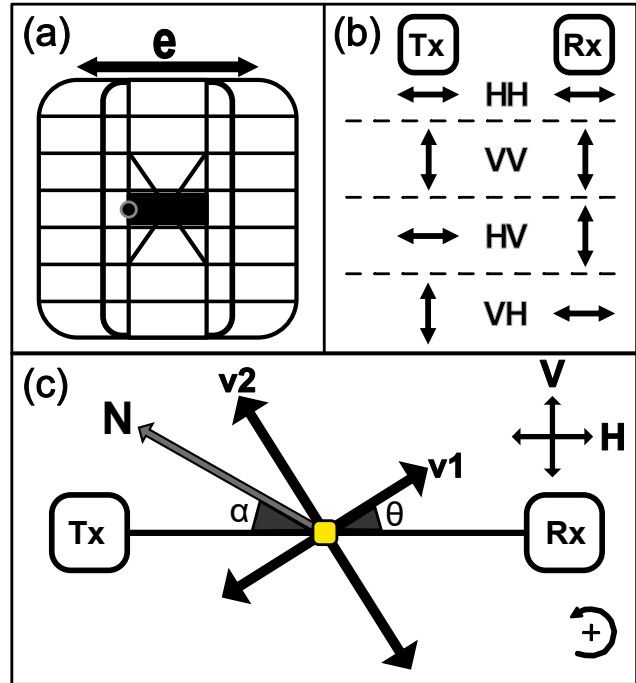

**Figure 2.** (a) Bird's eye view of the ApRES slot antenna with the direction of the electric field vector (**e**). (b) The terminology of the co-and cross-polarized ApRES measurements defined using **e**. The direction of wave propagation is into the page ($\otimes$). (c) The model coordinate system where transmitting (Tx) and receiving (Rx) antennas are connected with the aerial line (TR). The horizontal (H) and vertical (V) polarization planes are defined so that H is parallel to TR. **v1** and **v2** are the directions of the ice-fabric horizontal principal axes. $\theta$ is the angle between H and **v1**, and $\alpha$ is used for georeferencing.

The model coordinate system is shown in Fig. 2c. The aerial line (TR) connects transmitter (Tx) and receiver (Rx), and by convention, we assume that H is parallel to TR. **v1**, and **v2**, are the horizontal Eigenvectors which align with the direction of the smallest ($\epsilon'_x$) and largest ($\epsilon'_y$) horizontal principal permittivity, respectively (Fujita et al., 2006; Jordan et al., 2019). Hence, $\theta = 0°$ if H is aligned with **v1**. The angle $\alpha$ is measured by compass with $\pm15°$ uncertainty for georeferencing the data. Here, we use polar stereographic coordinates where anticlockwise rotation is positive.

Radar data at all the sites were collected at a fixed $\alpha$, obtained from different antenna orientation in co-polarization (HH, VV) and cross-polarization (HV, VH) configurations (Hargreaves, 1977; Fujita et al., 2006) as shown in Fig. 2b. We refer to these measurements as quad-polarimetric measurement. Radar data at Dome C were collected at 20 sites in January 2014. One of the sites is located within walking distance of the ice-core site EDC. The remaining 19 sites (termed E(ast)0-E18, and W(est)0.5-W18, with the numbers relating to the distance in km away from the dome) are aligned in a profile which is approximately

perpendicular to the long axis of the dome and parallel to the flowline (Fig. 1a). At EDML, data were collected in January 2017, approximately 2.7 km north-east of the ice-core site EDML (Fig. 1b). More information related to the individual ApRES sites are shown in Appendix A.





### 3.3 Background of radar polarimetry

Radio signal propagation through ice sheets is polarization-dependent because of the dielectric anisotropy of the ice fabric.

If the direction of $\mathbf{v3}$ is vertical and the remaining two Eigenvectors $(\mathbf{v1}, \mathbf{v2})$ are in the horizontal plane, then the relation between the depth profile of the dielectric permittivity tensor and the orientation tensor is given by Fujita et al. (2006):

$$\varepsilon'(z) = \begin{pmatrix} \epsilon'_x & 0 & 0 \\ 0 & \epsilon'_y & 0 \\ 0 & 0 & \epsilon'_z \end{pmatrix} = \begin{pmatrix} \epsilon'_\perp + \Delta\epsilon'\lambda 1 & 0 & 0 \\ 0 & \epsilon'_\perp + \Delta\epsilon'\lambda 2 & 0 \\ 0 & 0 & \epsilon'_\perp + \Delta\epsilon'\lambda 3 \end{pmatrix}. \tag{3}$$

For the dielectric permittivity at radio frequencies perpendicular to c-axes, we use $\epsilon'_\perp = 3.15$ (Fujita et al., 2000), which is slightly lower than the value found by Bohleber et al. (2012). The value of a dielectric anisotropy for a single crystal is set to

$\Delta\epsilon' \approx 0.034$ (Matsuoka et al., 1997). The vertical $\mathbf{v3}$ assumption in this study is justified through measurements at the EDC ice core where the direction of $\mathbf{v3}$ varies only by about 5° around the vertical (Durand et al., 2009). Elsewhere in ice sheets, this may not be the case, which will cause an additional source of horizontal birefringence (Matsuoka et al., 2009; Jordan et al., 2019).

We will model radio-wave propagation through birefringence ice using the method developed by Fujita et al. (2006). It

includes transmission and reflection of initially linearly polarized waves emitted with two polarization modes (H and V, with direction defined in the previous section). If $z$ is the depth from the surface (positive downward), it assumes stratified ice with $i = 1, ...N$ layers predicting the radar response as a function of the emitted polarization plane and ice-fabric parameters. Radar transmission ($\mathbf{T}$) and reflection ($\mathbf{\Gamma}$) are represented by $2 \times 2$ matrices only because radar signal propagation is insensitive to the vertically directed $\mathbf{v3}$. The transmitted radar wave ($\mathbf{e}_T$) and the corresponding radar reflection ($\mathbf{e}_R$) are $2 \times 1$ vectors, with

each component containing the electric field information of the H and V polarization components, respectively. Because only relative phase and amplitude variations are considered, all information about the radio wave transmission and reflection can be inferred from the scattering matrix ($\mathbf{S}$) at layer $N$:

$$\mathbf{e}_R = \mathbf{S}_N \mathbf{e}_T, \tag{4}$$

containing the complex scattering unit:

$$D(z) = \frac{\exp(jk_0 z)}{4\pi z}, \tag{5}$$

$$\mathbf{S}_N = \begin{pmatrix} s_{HH} & s_{VH} \\ s_{HV} & s_{VV} \end{pmatrix}_N = D^2(z_N) \prod_{i=1}^{N} [\mathbf{R}(\theta_{N+1-i})\mathbf{T}_{N+1-i}\mathbf{R}'(\theta_{N+1-i})]\mathbf{R}(\theta_i)\mathbf{\Gamma}_i\mathbf{R}'(\theta_i) \prod_{i=1}^{N} [\mathbf{R}(\theta_i)\mathbf{T}_i\mathbf{R}'(\theta_i)], \tag{6}$$

where $s_{HH}$ and $s_{VV}$ are the co-polarized scattering signals, $s_{HV}$ and $s_{VH}$ the cross-polarized scattering signals, j is the imaginary unit, and $k_0 = 2\pi f_c c_0^{-1}$ is the wavenumber in vacuum with $c_0$ the speed of light in vacuum. A rotation between TR and $\mathbf{v1}$ of the ice fabric in layer $i$, $(\theta_i)$, is accounted for by the rotation matrix $\mathbf{R}$ and its transpose ($\mathbf{R}'$)

$$\mathbf{R}(\theta_i) = \begin{pmatrix} \cos\theta_i & -\sin\theta_i \\ \sin\theta_i & \cos\theta_i \end{pmatrix}. \tag{7}$$





Transmission of the signal is described by the transmission matrix $\mathbf{T}$ along the ice-fabric horizontal principal axes. $\mathbf{T}$ is a function of wavenumbers ($k_x$, $k_y$), whereas the wavenumbers can be expressed as a function of dielectric permittivities ($\epsilon'_x$, $\epsilon'_y$) and electrical conductivities ($\sigma_x$, $\sigma_y$) (Fujita et al., 2006).

$$k_x = (\epsilon_0\mu_0\epsilon'_x\omega^2 + j\mu_0\sigma_x\omega)^{0.5}, \tag{8}$$

$$k_y = (\epsilon_0\mu_0\epsilon'_y\omega^2 + j\mu_0\sigma_y\omega)^{0.5}, \tag{9}$$

where $\epsilon_0$ and $\mu_0$ are the dielectric permittivity in vacuum and the magnetic permeability in vacuum, respectively, and $\omega$ is the angular frequency. In this study we follow Fujita et al. (2006) and assume isotropic electrical conductivity ($\sigma_x = \sigma_y$). Using Eq. (3), $\mathbf{T}$ can be written as a function of Eigenvalues:

$$\mathbf{T}(\lambda1_i, \lambda2_i) = \begin{pmatrix} T_x(\lambda1_i) & 0 \\ 0 & T_y(\lambda2_i) \end{pmatrix}, \tag{10}$$

where it tracks the relative phase shifts induced by the dielectric anisotropy along the ice-fabric principal axes. The reflection matrix $\mathbf{\Gamma}$ (Ulaby and Elachi, 1990) describes the reflection of the radio waves at an interface with changing dielectric properties

$$\mathbf{\Gamma}(\lambda1_i, \lambda2_i) = \begin{pmatrix} \Gamma_x(\lambda1_i) & 0 \\ 0 & \Gamma_y(\lambda2_i) \end{pmatrix}, \tag{11}$$

containing the Fresnel reflection coefficients $\Gamma_x$ and $\Gamma_y$ calculated from depth-variable changes in permittivity (Paren, 1981;
Ulaby and Elachi, 1990). In our case, these are caused by abrupt changes in ice-fabric orientation and/or magnitude. The reflection ratio $r = \frac{\Gamma_y}{\Gamma_x}$ is a measure for the polarization dependence of the reflection boundary. If anisotropic scattering is caused by depth variable ice fabric only, then the reflection ratio at the interfaces $i$ and $i+1$ can be approximated (Paren, 1981; Drews et al., 2012) by:

$$r_i = \left( \frac{\lambda2_i - \lambda2_{i+1}}{\lambda1_i - \lambda1_{i+1}} \right)^2. \tag{12}$$

This offers at least in theory the option to fully reconstruct $\lambda1$, $\lambda2$, and $\lambda3$ in a nadir geometry, which will resolve the ice-fabric types ambiguity as explained in Appendix B. Further details about the radar forward model implementation and definition of all the parameters in Eq. (6) are described in Fujita et al. (2006).

The parameters of interest that we aim to infer from the radar observations for each layer are the horizontal anisotropy $\Delta\lambda = \lambda2 - \lambda1$, the ice-fabric orientation angle $\theta$, and the reflection ratio $r$. All of these quantities may vary with depth. Much
information is gained by interpreting the phase difference between the ordinary and extraordinary radio waves along $\mathbf{v1}$ and $\mathbf{v2}$

$$\phi = \frac{4\pi f_c}{c_0} \int_z^0 (\sqrt{\epsilon'_x} - \sqrt{\epsilon'_y})dz + (\Delta\phi_x + \Delta\phi_y), \tag{13}$$





where $\Delta\phi_x$ and $\Delta\phi_y$ are the phase shift at the reflection boundary, which we set both to zero (Fujita et al., 2006; Jordan et al., 2019, 2020). The phase difference in Eq. (13) can vary rapidly with depth. Therefore, it is replaced by the coherence phase difference between $s_{HH}$ and $s_{VV}$, which is a crucial development in the works from Dall (2010). The coherence phase difference $\phi_{HHVV}$ is the argument of the complex polarimetric coherence $C_{HHVV}$, estimated via a discrete approximation,

$$C_{HHVV} = \frac{\sum_{b=1}^{M} s_{HH,b} \cdot s_{VV,b}^*}{\sqrt{\sum_{b=1}^{M} |s_{HH,b}|^2} \sqrt{\sum_{b=1}^{M} |s_{VV,b}|^2}}, \text{ with * as complex conjugate,} \tag{14}$$

$$\phi_{HHVV} = \arg(C_{HHVV}), \tag{15}$$

where $M$ is the number of range bins used for vertical averaging, and $b$ is the summation index. Using the depth gradient of $\phi_{HHVV}$ in a Taylor expansion of Eq. (13) termed scaled phase derivative,

$$\Psi = \frac{2c\sqrt{\epsilon'}}{4\pi f_c \Delta\epsilon'} \frac{d\phi_{HHVV}}{dz}, \tag{16}$$

which provides a way to relate the local phase gradient to $\Delta\lambda$ at the direction of the horizontal principal axes (Jordan et al., 2019, 2020)

$$\Delta\lambda(z) = \Psi(z, \theta = 0°, 90°). \tag{17}$$

The ApRES stores the de-ramped signal (Brennan et al., 2014; Jordan et al., 2020), which is not represented in Eqs. (14) and (15). The deramping corresponds to a complex conjugation of $C_{HHVV}$ (Jordan et al., 2020). Therefore, we use Eq. (14) for the models and the conjugate of Eq. (14) for the radar data to calculate the coherence phase.

We simplified Eq. (6) to a single layer case (Appendix C) showing that the polarity of $\Psi$ can differentiate the direction of **v1** and **v2** (Appendix D). If the coherence phase is based on the received signal, **v2** is in the direction of $\Psi > 0$ (TR ∥ **v2**), and **v1** is in the direction of $\Psi < 0$ (i.e., TR ∥ **v1**). When using observations, the depth gradient calculation of $\phi_{HHVV}$ is inherently difficult because any differencing scheme amplifies noise (Chartrand, 2011). We follow Jordan et al. (2019) and apply a 1D convolutional derivative, which also avoids phase unwrapping.

In Appendix E, we show that the quad-polarimetric measurement (Fig. 2c) can be used to synthesize the full radar return from any antenna orientation using a matrix transformation

$$\mathbf{S}_N(\theta \pm \gamma) = \mathbf{R}(\theta \pm \gamma)\mathbf{S}_N(\theta)\mathbf{R}'(\theta \pm \gamma), \tag{18}$$

where $\gamma$ is the angular offset from $\theta$. Equation (18) is the mathematical equivalent to rotating the antennas in the field for each polarimetric configuration. As demonstrated in Fig. E1, we find no significant differences between the synthesized and the full azimuthal rotation dataset with 22.5° increments. Hence, if **v3** is vertical a quad-polarimetric measurement is sufficient as it contains the full angular information.





### 3.4 Demonstration of anisotropic signatures in radar data using a synthetic model

For a given depth-profile of $\Delta\lambda(z)$, $\theta(z)$, and $r(z)$, the radar return can be simulated using the forward model described by Eqs. (4)-(6). We show a seven layers synthetic model in Fig. 3 to visualize features in the radar data, which can be linked to ice-fabric parameters. The model parameters used to generate Fig. 3 are shown in Table 2.

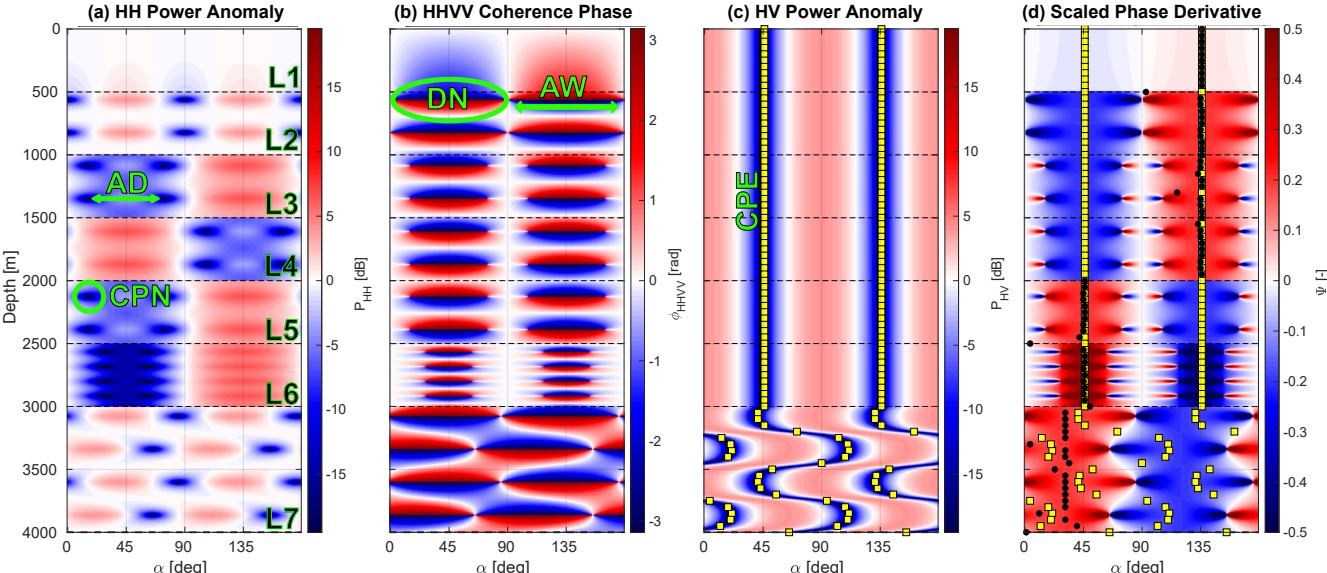

**Figure 3.** A seven layers synthetic model generated by Eq. (6) using the model parameters in Table (2). Horizontal black dashed lines are the layer boundaries with layer numbers from L1 to L7. (a) HH power anomaly ($P_{HH}$) representing co-polarization node (CPN) and node angular distance (AD). (b) HHVV coherence phase ($\phi_{HHVV}$) displaying dipole co-polarized node (DN) and node angular width (AW). (c) HV power anomaly ($P_{HV}$) representing cross-polarization extinction (CPE). (d) scaled phase gradient ($\Psi$) displaying the direction of **v1** (yellow squares in blue areas), and **v2** (yellow squares in red areas). The magnitude of $\Psi$ at the black dots is the value of $\Delta\lambda$.

Power anomalies illustrate the effects of anisotropic ice

$$P_{xx}(\theta, z) = 20 \log_{10} \left( \frac{|s_{xx}(\theta, z)|}{\frac{1}{n}\sum_{b=1}^{n} |s_{xx}(\theta_b, z)|} \right) \quad \text{for xx = HH, VV, HV,} \tag{19}$$

where $|s_{xx}|$ is the amplitude of the complex received signal, and $n$ is number of angular increments for $\theta$. In $P_{HH}$, a number of co-polarization nodes (CPN) occur, which result from destructive superposition of ordinary and extraordinary waves (Fig. 3a). The number of nodes per layer is only a function of ice fabric anisotropy in that layer, with higher horizontal anisotropy resulting in more nodes as predicted by Eq. (13). The nodes occur at a variable angular distance (termed AD in Fig. 3a) if anisotropic reflection is relevant (e.g., L2 and L3 in Fig. 3a). The angular dependency of the co-polarization nodes on anisotropic scattering can be identified using a depth-invariant ice-fabric orientation (constant $\theta$). Previously, Fujita et al. (2006) approximated the correlation between $AD$ and $r$ with a linear regression. As detailed in Appendix F we improved this





by finding the analytical solution

$$r = \frac{1}{\tan^2\left(\frac{AD}{2}\right)}. \tag{20}$$

Differences of both approaches are illustrated in Figure 4. Two important features in $P_{HH}$ are therefore the frequency of occurrence of co-polarization nodes with depth (a first-order proxy for the horizontal anisotropy) and their angular distance (a mixed proxy for anisotropic reflections or depth-variable ice-fabric orientation). $P_{HH}$ can be 90° (e.g., L2) or 180° (e.g., L3) symmetric if $r = 0$ dB or $r \neq 0$ dB, respectively.

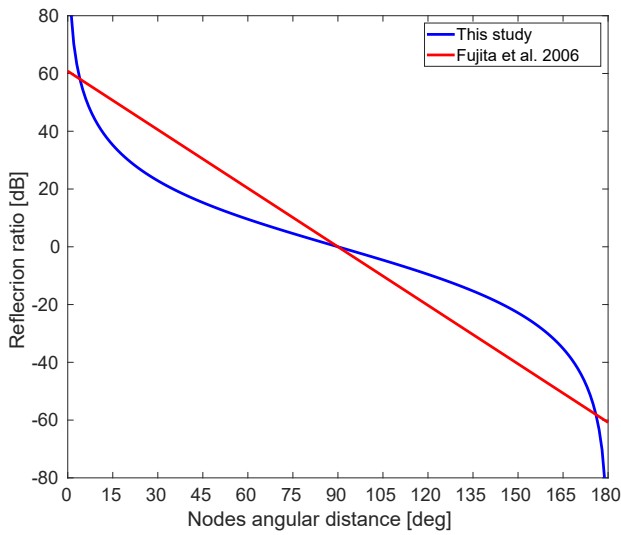

**Figure 4.** Dependence of reflection ratio on the azimuthal difference between two nodes as determined by Fujita et al. (2006) and through Eq. 20.

In a depth-invariant ice-fabric orientation, the minima in $P_{HV}$ align with **v1** and **v2** termed cross-polarization extinction

(CPE in Fig. 3c). Using the radar forward model, this can be derived analytically for a single layer case as:

$$P_{HV}(\theta, z) = 20 \log_{10}\left(\frac{\sin(\theta, z)\cos(\theta, z)}{\frac{1}{n}\sum_{b=1}^{n}\sin(\theta_b, z)\cos(\theta_b, z)}\right), \tag{21}$$

$$\theta = 0°, \quad \theta = \pm 90°. \tag{22}$$

In multi-layer cases, where $\theta$ changes with depth (e.g., L6 and L7 in Fig. 3b), $P_{HV}$ also depends on other parameters, making it difficult to infer $\theta$ using $P_{HV}$ alone.

The co-polarization nodes in $P_{HH}$ can also be observed in $\phi_{HHVV}$ (termed DN in Fig. 3b). The depth of the node can be automatically estimated at the zero-phase transition. Unlike $P_{HH}$, the nodes in $\phi_{HHVV}$ are 90° anti-symmetric, and their polarity is insensitive to $r$. This can be used to determine the directions of **v1** and **v2**. The angular width of the nodes (termed AW in Fig. 3b) decreases when $r \neq 0$ dB (e.g., L3 or L4). The absolute value of $\Psi$ at the principal axis's directions (**v1** or **v2**) is a first-order proxy for $\Delta\lambda$ at a given depth (Eq. 17, Fig. 3d).





**Table 2.** The model parameters used to generate Fig. 3. $\Delta\lambda$ is used in $\mathbf{T}$, assuming $\epsilon_x' = 3.15$, and $r$ is used in $\mathbf{\Gamma}$, assuming $\Gamma_x = 10^{-12}$. The vertical gridding of the model is 1 m.

| Layer Name | Depth [m] | $\Delta\lambda$ [−] | r [dB] | $\theta$ [°] |
|:---:|:---:|:---:|:---:|:---:|
| L1 | 0-500 | 0.025 | 0 | 45 |
| L2 | 500-1000 | 0.2 | 0 | 45 |
| L3 | 1000-1500 | 0.2 | 10 | 45 |
| L4 | 1500-2000 | 0.2 | -10 | 45 |
| L5 | 2000-2500 | 0.2 | -10 | 135 |
| L6 | 2500-3000 | 0.45 | -20 | 135 |
| L7 | 3000-4000 | 0.2 | 0 | 120 |

## 3.5 An inverse approach to infer ice fabric from quad-polarimetric returns

Fujita et al. (2006) focused on the power anomalies from co-and cross-polarized measurements ($P_{HH}$, $P_{HV}$). Dall (2010) and Jordan et al. (2019) included the coherence phase gradient ($\Psi$) to quantify the ice fabric horizontal anisotropy ($\Delta\lambda$). However, particularly for multi-layer cases where the ice-fabric parameters vary with depth, there has not yet been an established procedure for how ice-fabric parameters can be reliably inverted from observations. Here, we use the previous work from Fujita et al. (2006), Dall (2010), and Jordan et al. (2019) and provide additional justification to infer all the ice-fabric parameters in a continuous depth profile.

Our approach involves data preprocessing, initializing the model parameters, and parameter optimization using a constrained multivariable non-linear least-square inverse approach. All the three Eigenvalues are then estimated from the optimized $\Delta\lambda$ and $r$ using a top to bottom layer-by-layer approach assuming isotropic ice at the surface.

### 3.5.1 Data preprocessing

The full angular response is synthesized from HH, VV, and HV observations for a single TR orientation ($\theta$) using Eq. (18) at 1° increments. The amount and method of smoothing data depend on nodes' vertical frequency and phase polarity's sharpness. The power anomalies are smoothed by moving average and 2D Gaussian convolution. The coherence phase ($\phi_{HHVV}$) is inherently smoothed, depending on the size of the depth window in Eq. (14), while its gradient ($\Psi$) is smoothed with a 1D Gaussian convolution at each azimuth.

### 3.5.2 Model parameterization

We investigate two parameterization types for the free model parameters ($\theta$, $\Delta\lambda$, $r$) with depth: piece-wise constant and a superposition of Legendre Polynomials. The former has the highest number of free model parameters but can capture abrupt variability with depth. The latter has a reduced set of free model parameters with improved performance during the inversion, but varies more smoothly with depth. At Dome C, no abrupt variability is visible in the data so that we use the Legendre



Polynomials with 50 free model parameters (30 for $\theta$, 10 for $\Delta\lambda$, and 10 for $r$). At EDML, we default to the piecewise constant parameterization resulting in 120 free model parameters (40 piecewise constant intervals at 50 m spacings for each model parameter).

### 3.5.3 Derivation of initial guess

The non-linear optimization problem depends on a well-defined initial guess based on our inferences from the synthetic data. Initial guesses of variables are marked with superscript 0. We first derive the initial guess for the orientation of the ice fabric ($\theta^0(z)$) using the minima in $P_{HV}$, polarity in $\phi_{HHVV}$, and the sign of $\Psi$. We then infer $\Delta\lambda^0(z)$ using the absolute value of $\Psi$ at the minima of $P_{HV}$. The initial guess for $r^0(z)$ is zero. The underlying assumption for all of the initial guesses is that $\theta$ does not vary significantly with depth.

### 255 3.5.4 Cost function and optimization

We optimize $\Delta\lambda$, $\theta$, and $r$ for all depth intervals. There are a number of possible model data misfit metrics of power anomalies and phase differences

$$J_{\phi_{HHVV}} = ||\phi_{HHVV}^{\text{obs.}} - \phi_{HHVV}^{\text{mod.}}||^2, \tag{23}$$

$$J_{P_{HH}} = ||P_{HH}^{\text{obs.}} - P_{HH}^{\text{mod.}}||^2, \tag{24}$$

$$J_{P_{HV}} = ||P_{HV}^{\text{obs.}} - P_{HV}^{\text{mod.}}||^2, \tag{25}$$

and the total misfit between the observed (obs.) and the modeled data (mod.) is defined as:

$$J_{total} = l_1(J_{\phi_{HHVV}}) + l_2(J_{P_{HH}}) + l_3(J_{P_{HV}}), \tag{26}$$

where $l_1$, $l_2$, and $l_3$ are constants. In Table 3, we show the values of $l_1$, $l_2$, and $l_3$ that we used for Dome C and EDML sites. Using $l_1$ (for coherence phase misfit) in EDML is not applicable because the phase is too noisy. Therefore, we did not 265 optimize $\Delta\lambda$ for this site. To further constrain the inversion, we set bounds on the model parameters so that $0 < \Delta\lambda_i < 0.5$, $0° < \theta_i < 180°$, and $-30$ dB $< r_i < 30$ dB. This is implement in the cost function in the form of log-barrier functions using Matlab®'s fmincon algorithm.

**Table 3.** The constant $l_1$, $l_2$, $l_3$ for each ice-fabric parameter at Dome C and EDML

| Site | $\theta$ | $\Delta\lambda$ | $r$ |
|------|------|------|------|
| Dome C | 1,0,0 | 1,0,0 | 0,1,0 |
| EDML | 0,1,1 | 0,0,0 | 0,1,1 |





### 3.6 Reconstruction of all Eigenvalues

Once the radar forward model is optimized, we attempt to reconstruct all the three Eigenvalues in a top-to-bottom approach by solving Eq. 12 using the optimized $r$ and $\Delta\lambda$. At the surface ice is isotropic so that $\lambda 1_1 \approx 0.33$ allowing to infer $\lambda 2_1$ and $\lambda 3_1$ from $\Delta\lambda_1$

$$\lambda 2_1 = \Delta\lambda_1 + \lambda 1_1, \tag{27}$$

$$\lambda 3_1 = 1 - \lambda 2_1 + \lambda 1_1. \tag{28}$$

For deeper layers $i+1$, all three Eigenvalues, can in theory, be reconstructed analytically by solving

$$r_i = \left( \frac{(\Delta\lambda_{i+1} - \Delta\lambda_i) + (\lambda 1_{i+1} - \lambda 1_i)}{\lambda 1_{i+1} - \lambda 1_i} \right)^2, \tag{29}$$

for $\lambda 1_{i+1}$ and infer $\lambda 2_{i+1}$ and $\lambda 3_{i+1}$ with

$$\lambda 2_{i+1} = \Delta\lambda_{i+1} + \lambda 1_{i+1}, \tag{30}$$

$$\lambda 3_{i+1} = 1 - \lambda 2_{i+1} + \lambda 1_{i+1}. \tag{31}$$

However, errors during the optimization may result in a reconstruction of the three Eigenvalues, which do not comply with limits inferred in Sect. 3.1. In that case, $\Delta\lambda$ and $r$ are varied in a systematic search to find Eigenvalues within the permissible limits. Solutions, in this case, are not unique, and additional constraints on the vertical gradients (here: $1.0 \cdot 10^{-6} < \frac{d\lambda 3}{dz} < 1.5 \cdot 10^{-3}$) are required. This correction does not significantly alter the results from the previous section but assures that the inferred Eigenvalues are internally consistent.

## 4 Results

### 4.1 Ice-fabric parameters from polarimetric ApRES at Dome C

Polarimetric ApRES data collected at Dome C is shown in Figs. 5a-d. A co-polarization node occurs at 1100 m depth, and a second node develops at about 2000 m depth (Figs. 5a, b). The existence of only one pair of nodes over 2000 m indicates comparatively small horizontal ice anisotropy (i.e., low $\Delta\lambda$) similar to what has been observed at Dome Fuji (Fujita et al., 2006). The angular distance between the two co-polarization nodes is close to 90°, consistent with $r$ close to 0 dB (Fig. 5a). $P_{HV}$ shows little depth-variability (Fig. 5c), suggesting that the ice-fabric orientation angle ($\theta$) does not vary strongly with depth. The scaled phase derivative ($\Psi$, Fig. 5d) is unclear in terms of polarity for the top 150 m. Below that, the polarity more clearly indicates the orientation of the largest horizontal Eigenvectors.

Optimized model results in Figs. 5e-h reproduce the principal patterns of the radar observations. The reconstructed Eigenvalues (Fig. 5i) capture the observed transition from isotropic to a girdle-type ice-fabric in the ice-core data. The reconstructed horizontal anisotropy (Fig. 5j) captures the mean well ($\overline{\Delta\lambda}_{(z>150\text{m})} = 0.037$), albeit showing less depth variability than the





observations. The ice-fabric orientation at the top 150 m is poorly constrained due to the low horizontal anisotropy (Fig. 5k). The mean orientation of **v2** below 150 m is 124° relative to True North, which is almost perpendicular to the surface flow direction towards 45°. The orientation cannot be validated with ice-core data, which is azimuthally unconstrained. The mean estimated reflection ratio below 150 m is low ($\overline{r}_{(z>150\mathrm{m})} = -3$ dB, Fig. 5l), indicating that the role of anisotropic reflections

is small.

**Figure 5.** Results for EDC: (a)-(d) radar observations with green lines in (c) and (d) marking the minima in $P_{HV}$. (e)-(h) optimized model output capturing the principle patterns of the observations. (i)-(l) inferred model parameters validated with ice-core data (Durand et al., 2009) in terms of Eigenvalues (i) and horizontal anisotropy (j). The inferred **v2** is perpendicular to the mean surface flow direction (k), and the anisotropic reflection ratio is small (l). Note that the negative $\Psi$ in (d) is masked for a better demonstration of **v2** orientation.





## 4.2   Ice-fabric parameters from polarimetric ApRES at EDML

Next, we apply our method to ApRES data collected at the EDML drill site. Contrary to what has been observed at Dome C, co-polarization nodes can barely be localized in $P_{HH}$ as no 90° symmetry is apparent (Fig. 6a). This indicates that anisotropic scattering is relevant ($r \neq 0$ dB), as already noticed earlier (Drews et al., 2012). Moreover, the coherence phase shows many

nodes (Fig. 6b), indicating a much stronger horizontal anisotropy (i.e., large $\Delta\lambda$). This is comparable to the ice core at Mizuho, equally located in a flank flow regime (Fujita et al., 2006). Although $P_{HV}$ shows almost no depth variability in ice-fabric orientation (Fig. 6c), it is not straightforward to identify the direction of **v1** and **v2** using the polarity of $\Psi$ because of the strong ice anisotropy (Fig. 6d).

The optimized model (Figs. 6e-h) reproduces all basic features seen in the radar data. Inferred model parameters closely

follow the ice-core measurements both in terms of absolute Eigenvalues (Fig. 6i) and horizontal anisotropy (Fig. 6j). The shallower development of the girdle ice fabric compared to Dome C is detected. The mean estimated horizontal anisotropy below 200 m in EDML ($\overline{\Delta\lambda}_{(z>200\text{m})} = 0.265$) is more than seven times stronger than Dome C. The mean inferred orientation of **v2** below 200 m is 174° relative to True North (Fig. 6k). Similar to Dome C, this is near perpendicular to the ice-flow direction at the surface towards 90°. The estimated reflection ratio in EDML (Fig. 6l) can be divided into two major zones

($\overline{r}_{(200\text{m}<z<850\text{m})} = 16$ dB, and $\overline{r}_{(z>850\text{m})} = -15$ dB). Contrary to Dome C, anisotropic reflections are more relevant, and the previously suggested existence of two anisotropic scattering zones above and below approx. 850 m (Drews et al., 2012) appears in the observations and the optimized model output.





**Figure 6.** Results for EDML: same as Fig. 5, with the exception of the measured parameters in **i** and **j** are from Weikusat et al. (2017).

## 4.3 Spatial variability of ice-fabric parameters in the local dome-flank transition zone

After investigating specific characteristics of a dome position (EDC) and a flank flow regime (EDML), we next investigate a
local dome-to-flank transition (36 km). At Dome C, 19 sites are located along a profile extending 18 km away to either side
from the local ice dome (Fig. 1a), and a summary of the results is presented in Fig. 7. We focus on the upper 2000 m, where
the signal to noise ratio is sufficiently high. All stations yield coherent results showing an isotropic ice fabric that gradually
evolves into a weak girdle with depth. The depths of the first co-polarization nodes can be detected at all sites (green-dashed
line in Fig. 7b). It is shallowest beneath the dome and moves to larger depths further away from the dome in the flanks. The
depth-variability of the co-polarization nodes results in a $\Delta\lambda$ that is most anisotropic beneath the dome, and less anisotropic

in the flanks (Fig. 7c). The orientation of the Eigenvectors is poorly constrained in the upper 200 m. At larger depths, they are oriented parallel (**v1**) and perpendicular (**v2**) to the surface flow direction in-line with what has been inferred in Sect. 4.1.

**Figure 7.** Ice-fabric evolution in the local dome-to-flank transition at Dome C. (a) surface (Howat et al., 2019) and bed (Morlighem et al., 2020) elevation in meter above sea level. Red crosses are the measured bed elevation from radar power at each site. (b) observed polarimetric coherence phase difference ($\phi_{HHVV}$) at each site. The green dashed line connects the nodes at each site. (c) the optimized horizontal anisotropy ($\Delta\lambda$). (d) the optimized orientation of the largest horizontal Eigenvector (v2). The red rectangle in the legend marks the surface flow direction. All panels are corrected for the surface elevation.



## 5 Discussion

### 5.1 Radar polarimetry as a tool to characterize ice-fabric variability horizontally and vertically

Our method extracts the horizontal anisotropy, orientation, and the anisotropic reflection ratio of the ice fabric as a function of depth. We also estimate all three Eigenvalues required for the dielectric orientation tensor. This can be compared with laboratory measurements (Durand et al., 2009) and integrated into ice-flow models (Gagliardini et al., 2009). Our main assumption is that the strongest Eigenvector (and with it the orientation tensor) is aligned in the vertical. We now discuss the limitations of our approach.

In terms of the data pre-processing, there are no structural differences in our data between synthesizing the polarization dependency out of a single set of quad-polarimetric measurement (Appendix E) and the more common polarimetric measurements in glaciology where antennas are kept parallel or perpendicular while being rotated several increments between 0 and 180 degrees (Fujita et al., 2006). However, more systematic investigation is warranted if this also holds when the ice fabric is not aligned vertically.

The signal quality and noise level, particularly in the HHVV coherence phase, are important. In areas with high horizontal anisotropy and consequently densely spaced co- and cross-polarization nodes (i.e., the EDML case), care needs to be taken that the denoising does not average over multiple nodes. Derivation of the initial guess for the inverse approach depends on the data quality and is guided by characteristic features in synthetic forward models, some of which can be analytically described for one layer cases. Multi-layer cases, however, are difficult to interpret, particularly if the ice-fabric orientation changes strongly 345 (by several 10s of degrees) with depth. Fortunately, this does not appear to be the case for the data presented here, so that the initial guess already results in a forward model that adequately captures characteristic features in the data. The optimization improves the model–data misfit but does not lead to significant differences with our first informed guess. Nevertheless, this step is required to predict the depth-variability of all the three Eigenvalues.

The reconstruction of the Eigenvalues assumes isotropic ice/firn at the surface. This is reasonable for the dome and flank-flow 350 settings considered here, but may need to be revisited in other settings where fabric can develop near the surface as ice-streams and ice-shelves. More critical is the reflection ratio itself, which is ill-constrained in magnitude and amplifies small changes in the Eigenvalues across the reflection boundaries. This is mitigated by the range of allowed Eigenvalues (Sect. 3.1), and it is those constraints that facilitate the derivation of all Eigenvalues from the anisotropic reflection ratio. The predicted Eigenvalues ($\lambda 1$, $\lambda 2$, and $\lambda 3$) in this method such show a good match to the ice-core observations in both cases.

The azimuthal constraints that radar polarimetry provides can, in general, not be validated by ice-core measurements with few exceptions (e.g., Westhoff et al., 2020). However, the alignment of the ice-fabric principal axes with the surface-flow direction detailed below adds credibility to our inferences and shows advantages of this approach over previous attempts focusing on the power anomalies only (Fujita et al., 2006; Matsuoka et al., 2012). The underlying reason for this is that the polarity of the depth gradient of the polarimetric coherence phase is independent of anisotropic scattering.

The inversion requires an initial guess (Sect. 3.5.3) that is based on experience from synthetic test cases. In our experience with radar polarimetry and the explored ice dynamic context, this grants a robust solution, also because a wrong initial guess





results in a large model-data misfit that can be identified easily. In the future, this can be improved by using gradient-free optimization schemes (e.g., in a Bayesian framework) that can correct for a poor initial guess by exploring the parameter space more systematically.

Our strongest assumption is that the strongest Eigenvector (**v3**) should be close to vertical. While this assumption is justified here, as flow at domes is dominated by vertical compression and crystal c-axis tend to align in vertical, it may not apply elsewhere in ice sheets and cause an additional source of horizontal birefringence (Matsuoka et al., 2009). While it is possible to explore the effects of other than the largest Eigenvector being vertical (Jordan et al., 2019, p. 13), it is impossible to circumvent that the radio-wave propagation is vertical and hence insensitive to changes along that direction. In the future, we envision the

use of wide-angle surveys with curved ray paths (e.g., Winebrenner et al., 2003) to overcome this limitation.

## 5.2    Spatial variability of ice-fabric types in a dome-flank transitions

We now turn to the ice-dynamic context of our inferred characteristics of ice-fabric variation from dome, where flow is dominated by vertical compression to flank flow regimes, where flow is driven by vertical shear. Our inverse approach shows higher horizontal ice anisotropy at EDML compared to Dome C throughout the ice column. This increase from the dome to the flank

supports earlier inferences that ice anisotropy is larger in areas with significant horizontal strain compared to settings where vertical compression is dominant (Fujita et al., 2006; Matsuoka et al., 2012). This is in contrast, however, with the observed decrease in ice anisotropy in the Dome C transect (Fig. 7c), where the ice fabric is more anisotropic at the Dome compared to the flanks. Our hypothesis is that this near-field anomaly reflects ice-dynamic modification of ice fabric through the Raymond effect (Raymond, 1983). Martín et al. (2009) predict local, ice-dynamically induced ice-fabric variability up to approximately

5 ice thickness to either side of the ice divides. The 36 km long Dome C transect images an ice thickness of about 3000 m and hence approximately covers this domain. The absence of Raymond arches in the radar stratigraphy beneath Dome-C (Cavitte et al., 2016, p. 325) suggests that these need a longer time to evolve, whereas the ice-fabric pattern reflects the instantaneous operation of the Raymond effect. We acknowledge that there are other explanations for the ice-fabric pattern under Dome C, such as across-profile flow or bedrock influence. In any case, we want to highlight here how, due to the spatial extension of

our observations, our inferred fabric distributions combined with an anisotropic flow model can be used to test these and other hypothesis.

Both in EDML and Dome C areas, the inferred ice-fabric orientation varies little over the depth-intervals considered, and in both cases, the inferred orientations line-up with the surface flowfield. More specifically, **v1** is approximately oriented along-flow and **v2** is approximately oriented across-flow. Those directions also align with the principal strain rate components

in Dome C (Rémy and Tabacco, 2000; Vittuari et al., 2004) and EDML (Drews et al., 2012) (Fig. 1). In both cases, **v2** is approximately parallel to the direction of the maximal principal strain-rate component, whereas **v1** is aligned with along-flow minimal principal strain-rate component. (At Dome C, where flow velocities are small, derivation of the strain-rate field is not trivial and builds on additional assumptions of the surface topography).

More theoretical work is required to understand the vertical variability in horizontal anisotropy, which is picked up in radar

polarimetry through the strength of the anisotropic reflection ratio. At EDML, the reflection ratio is a dominant and required





factor to explain the radar signatures, while at Dome C, it is close to negligible. Fujita et al. (2006) have observed a similar increase in anisotropic scattering between Dome-F and Mizuho, suggesting that this may be a generic feature in ice sheets that requires more investigation. Contrary to EDML, the signal at Dome C is dominated by birefringence, and the contribution of anisotropic reflection is small. Yet, it appears that it leaves a small signature in the data that can be exploited. Moreover,

our analysis suggests that there are no other mechanisms (e.g., a directional interface roughness) contributing to anisotropic reflections. This point requires confirmation from other ice-core sites because the recovery of all three Eigenvalues (and their corresponding directions) offers significant possibilities to constrain ice fabric in ice sheets in general.

## 6    Conclusions

We show here, for the first time, the spatial distribution of ice-fabric in domes: from the summit, where flow is dominated by

vertical compression, to the flanks, where flow is driven by vertical shear. The combination of co- and cross-polarized power anomaly along with the depth gradient of polarimetric coherence phase provides three major parameters and their changes over depth, i.e., the ice-fabric orientation, horizontal anisotropy, and its vertical variability. We quantify these changes using an inverse approach that extracts ice-fabric information from radar polarimetry. We present here a method to combine them and infer the full orientation tensor. We validate our technique with data from two ice-core locations situated in contrasting

ice-flow regimes. The inferred ice-fabric orientation aligns with the observed surface velocity and surface strain rate fields. This suggests that polarimetric radar is an ideal tool to map ice-fabric characteristics elsewhere as well.

We present ice-fabric spatial distribution across a flow-plane at Dome C. The 20 ApRES sites in that area are internally consistent, and small changes in the horizontal anisotropy can horizontally be tracked in the polarimetric coherence phase. We detect a minor decrease in horizontal anisotropy away from the dome that we tentatively link to the operation of the Raymond

effect. On larger spatial scales, the horizontal anisotropy increases in the flanks (i.e., at EDML), and our findings are consistent with previous studies. Our analysis suggests that ice-fabric characteristics can now be reliably inferred in larger parts of the Antarctica and Greenland ice sheet, given that more and more profiles are recorded in coherent and in the quad-polarimetric configuration. This will be a decisive step to further constrain the anisotropic nature of ice and understand better its contribution to internal deformation.

*Code and data availability.*    Codes related to this study are available on Github (https://github.com/RezaErshadi/ApRES_InverseApproach.git) under the GNU GPLv3 license. Radar data at EDML (Christmann et al., 2020) can be found on Pangaea (https://www.gnu.org/licenses/gpl-3.0.en.html). Radar data at Dome C will be provided on request to the authors.





## Appendix A: ApRES stations info table

**Table A1.** ApRES stations info. Coordinates are shown in decimal degrees in the WGS84 reference system. Surface elevations are based on REMA (Helm et al., 2014). Bed elevations are obtained from the polarimetric radar data. Tx-Rx azimuth is measured by a compass with $\pm 15°$ tolerance.

| Site Name | Location | Longitude [DD] | Latitude [DD] | Surface elevation [m asl] | Bed elevation [m asl] | Tx-Rx azimuth [°] |
|---|---|---|---|---|---|---|
| LD01 | EDML | 0.093410 | -74.995730 | 2892.3 | 206.5 | 114 |
| EPICA | Dome C | 123.350000 | -75.100000 | 3232.7 | -8.0 | 163.6 |
| W18 | Dome C | 122.909370 | -75.000790 | 3226.9 | -119.28 | 81.2 |
| W12 | Dome C | 123.071950 | -75.035100 | 3229.0 | 64.5 | 64.3 |
| W06 | Dome C | 123.237540 | -75.068530 | 3232.4 | 26.0 | 76.2 |
| W4d5 | Dome C | 123.280150 | -75.076690 | 3233.1 | 24.4 | 69 |
| W2d5 | Dome C | 123.337480 | -75.086960 | 3233.5 | 24.8 | 62.2 |
| W1d5 | Dome C | 123.366290 | -75.092090 | 3233.5 | 51.4 | 69.3 |
| W1d0 | Dome C | 123.381070 | -75.094670 | 3233.6 | 64.7 | 71.9 |
| W0d5 | Dome C | 123.395540 | -75.097190 | 3233.5 | 54.45 | 75.6 |
| E0 | Dome C | 123.410151 | -75.099738 | 3233.7 | 36.6 | 71.5 |
| E0d5 | Dome C | 123.424700 | -75.102290 | 3233.5 | 50.5 | 67.8 |
| E1d0 | Dome C | 123.439460 | -75.104780 | 3233.5 | 80.6 | 61.7 |
| E1d5 | Dome C | 123.453870 | -75.107310 | 3233.3 | 109.2 | 64.5 |
| E02 | Dome C | 123.468390 | -75.109810 | 3233.1 | 121.5 | 73.3 |
| E03 | Dome C | 123.497900 | -75.114910 | 3232.8 | 78.0 | 71.9 |
| E4d5 | Dome C | 123.541160 | -75.122690 | 3232.27 | 116.4 | 65.8 |
| E06 | Dome C | 123.583990 | -75.131010 | 3231.3 | 38.0 | 58.5 |
| E09 | Dome C | 123.666480 | -75.147581 | 3229.1 | 38.1 | 61.4 |
| E12 | Dome C | 123.748400 | -75.164990 | 3227.2 | 50.3 | 57.8 |
| E18 | Dome C | 123.906540; | -75.201260 | 3224.8 | 17.8 | 70.2 |

## Appendix B: The effect of vertical insensitivity in polarimetric radar

Since polarimetric radar is insensitive to the vertical component of ice fabric, it is only possible to estimate its horizontal anisotropy (Sect. 3.3). As shown in Fig. B1, the value of $\Delta\lambda = \lambda 2 - \lambda 1$ is not sufficient to infer the ice-fabric type. End-member cases in Figs. B1a-c are the values for $\lambda 1$, $\lambda 2$, and $\lambda 3$ for an isotropic (I), single-pole maximum (S), and girdle type (G) ice fabric. Although, the uncertainty in detecting the ice-fabric type decreases for stronger $\Delta\lambda$, to constrain the ice-fabric type from the polarimetric radar, all three Eigenvalues along the ice-fabric principal axes are necessary.





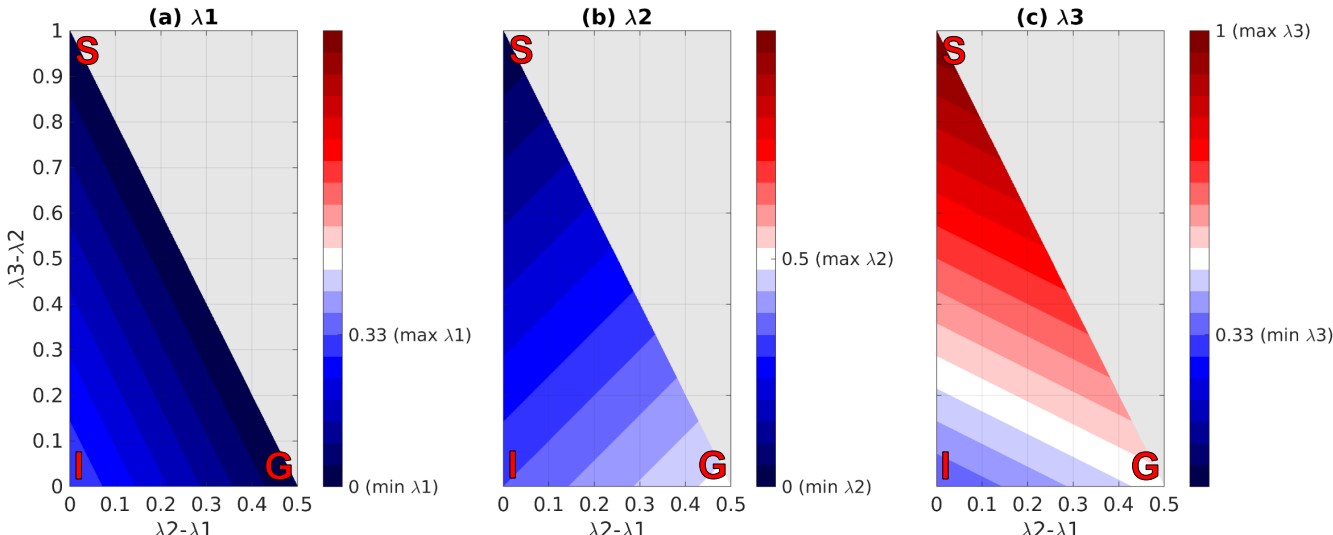

**Figure B1.** Ice fabric type and Eigenvalue (a)$\lambda 1$, (b)$\lambda 2$, (c)$\lambda 3$ as a function of Eigenvalue differences $\lambda 2 - \lambda 1$ and $\lambda 3 - \lambda 2$. (I) isotropic ice fabric where $\lambda 2 - \lambda 1$ and $\lambda 3 - \lambda 2 = 0$. (S) single-pole maximum ice fabric where $\lambda 2 - \lambda 1 = 0$ and $\lambda 3 - \lambda 2 = 1$. (G) vertical girdle ice fabric where $\lambda 2 - \lambda 1 = 0.5$ and $\lambda 3 - \lambda 2 = 0$.

**Appendix C: Matrix-based radio wave propagation in a single layer case**

Here we expand individual components of a single layer case that are used later to determine the relationship between the anisotropic reflection ratio and the angular distance of the co-polarization nodes. For this case, we drop the indices relating to the different layers and expand Eq. (6):

$$\mathbf{S} = D^2 \mathbf{R}(\theta) \mathbf{T}^2 \mathbf{\Gamma} \mathbf{R}'(\theta), \tag{C1}$$

$$\mathbf{S} = \begin{pmatrix} s_{HH} & s_{VH} \\ s_{HV} & s_{VV} \end{pmatrix} = D^2 \begin{pmatrix} T_x^2 \Gamma_x \cos^2 \theta + T_y^2 \Gamma_y \sin^2 \theta & \sin \theta \cos \theta (T_x^2 \Gamma_x - T_y^2 \Gamma_y) \\ \sin \theta \cos \theta (T_x^2 \Gamma_x - T_y^2 \Gamma_y) & T_y^2 \Gamma_y \cos^2 \theta + T_x^2 \Gamma_x \sin^2 \theta \end{pmatrix}. \tag{C2}$$

so that:

$$s_{HH}(\theta \pm \frac{\pi}{2}) = s_{VV}(\theta), \tag{C3}$$

$$s_{HV}(\theta \pm \frac{\pi}{2}) = -s_{HV}(\theta). \tag{C4}$$





The complex $s_{HH}$, its amplitude, and its phase are then:

$$s_{HH} = \frac{1}{(4\pi z_1)^2} \left( \Gamma_x \cos^2(\theta) \exp(j2zk_x) + \Gamma_y \sin^2(\theta) \exp(j2zk_y) \right), \tag{C5}$$

$$|s_{HH}| = \frac{\Gamma_x}{(4\pi z)^2} \left( \cos^4(\theta) + r^2 \sin^4(\theta) + 2r \sin^2(\theta) \cos^2(\theta) \cos(2z(k_x - k_y)) \right)^{0.5}, \tag{C6}$$

$$\arg(s_{HH}) = \tan^{-1} \left( \frac{\sin(2zk_x) + r \tan^2(\theta) \sin(2zk_y)}{\cos(2zk_x) + r \tan^2(\theta) \cos(2zk_y)} \right), \tag{C7}$$

respectively. Also, the complex $s_{HV}$, its amplitude, and its phase, respectively:

$$s_{HV} = \frac{\sin(\theta)\cos(\theta)}{(4\pi z)^2} \left( \Gamma_x \exp(j2zk_x) - \Gamma_y \exp(j2zk_y) \right), \tag{C8}$$

$$|s_{Hv}| = \frac{\Gamma_x \sin(\theta)\cos(\theta)}{(4\pi z)^2} \left( 1 + r^2 - 2r \cos(2z(k_x - ky)) \right)^{0.5}, \tag{C9}$$

$$\arg(s_{HV}) = \tan^{-1} \left( \frac{\sin(2zk_x) + r \sin(2zk_y)}{\cos(2zk_x) + r \cos(2zk_y)} \right). \tag{C10}$$

## Appendix D: Polarity of the coherence phase gradient

This section details the relationship between the polarity of the phase gradient and the corresponding directions of the Eigenvectors. Care has to be taken here, as the de-ramping during ApRES data acquisition is equivalent to a complex conjugation of the received signal. If this is not accounted for, the inferred Eigenvector **v1** and **v2** will be swapped. More specifically, for a received signal at $\theta = 0°$:

$$s_{HH} = A \left( \Gamma_x \cos(2zk_x) + j\Gamma_x \sin(2zk_x) \right), \tag{D1}$$

$$s_{VV} = A \left( \Gamma_y \cos(2zk_y) + j\Gamma_y \sin(2zk_y) \right), \tag{D2}$$

so that the coherence phase results in:

$$C_{HHVV} = \left( \cos(2z(k_x - k_y)) + j \sin(2z(k_x - k_y)) \right), \tag{D3}$$

$$\phi_{HHVV}(\theta = 0) = 2z(k_x - k_y), \tag{D4}$$

and conversely for $\theta = 90°$:

$$\phi_{HHVV}(\theta = 90°) = 2z(k_y - k_x). \tag{D5}$$

As explained in Sect. 3.3, $k_x$ and $k_y$ are a function of $\lambda 1$ and $\lambda 2$, respectively. Because $\lambda 1 \leq \lambda 2$ it follows that $k_x < k_y$. Therefore, $\phi_{HHVV}(\theta = 0°) < 0$ and $\frac{\phi_{HHVV}(\theta = 0°)}{dz} < 0$. The reverse holds for $\theta = 90°$.





## Appendix E:  Reconstruction of azimuthal measurements from a single quad-polarimetric acquisition

The transformation is purely geometrical and corresponds to a coordinate transformation into a rotated reference system for an arbitrary $\gamma$:

$$
\begin{pmatrix} s_{HH}(\theta \pm \gamma) & s_{VH}(\theta \pm \gamma) \\ s_{HV}(\theta \pm \gamma) & s_{VV}(\theta \pm \gamma) \end{pmatrix} = \begin{pmatrix} \cos(\theta \pm \gamma) & -\sin(\theta \pm \gamma) \\ \sin(\theta \pm \gamma) & \cos(\theta \pm \gamma) \end{pmatrix} \begin{pmatrix} s_{HH}(\theta) & s_{VH}(\theta) \\ s_{HV}(\theta) & s_{VV}(\theta) \end{pmatrix} \begin{pmatrix} \cos(\theta \pm \gamma) & \sin(\theta \pm \gamma) \\ -\sin(\theta \pm \gamma) & \cos(\theta \pm \gamma) \end{pmatrix}, \quad \text{(E1)}
$$

resulting in:

$$
s_{HH}(\theta \pm \gamma) = \cos^2(\theta \pm \gamma)s_{HH}(\theta) + \sin^2(\theta \pm \gamma)s_{VV}(\theta) - \sin(\theta \pm \gamma)\cos(\theta \pm \gamma)(s_{HV}(\theta) + s_{VH}(\theta)), \quad \text{(E2)}
$$

$$
s_{VH}(\theta \pm \gamma) = \cos^2(\theta \pm \gamma)s_{VH}(\theta) - \sin^2(\theta \pm \gamma)s_{HV}(\theta) + \sin(\theta \pm \gamma)\cos(\theta \pm \gamma)(s_{HH}(\theta) + s_{VV}(\theta)), \quad \text{(E3)}
$$

$$
s_{HV}(\theta \pm \gamma) = \cos^2(\theta \pm \gamma)s_{HV}(\theta) - \sin^2(\theta \pm \gamma)s_{VH}(\theta) + \sin(\theta \pm \gamma)\cos(\theta \pm \gamma)(s_{HH}(\theta) + s_{VV}(\theta)), \quad \text{(E4)}
$$

$$
s_{VV}(\theta \pm \gamma) = \cos^2(\theta \pm \gamma)s_{HH}(\theta) + \sin^2(\theta \pm \gamma)s_{VV}(\theta) + \sin(\theta \pm \gamma)\cos(\theta \pm \gamma)(s_{HV}(\theta) + s_{VH}(\theta)). \quad \text{(E5)}
$$

Figure E1 demonstrates this approach for EDML site, where quad-polarimetric measurements were additionally complemented with a dataset collected with rotating antennas. There are no structural differences between both datasets.



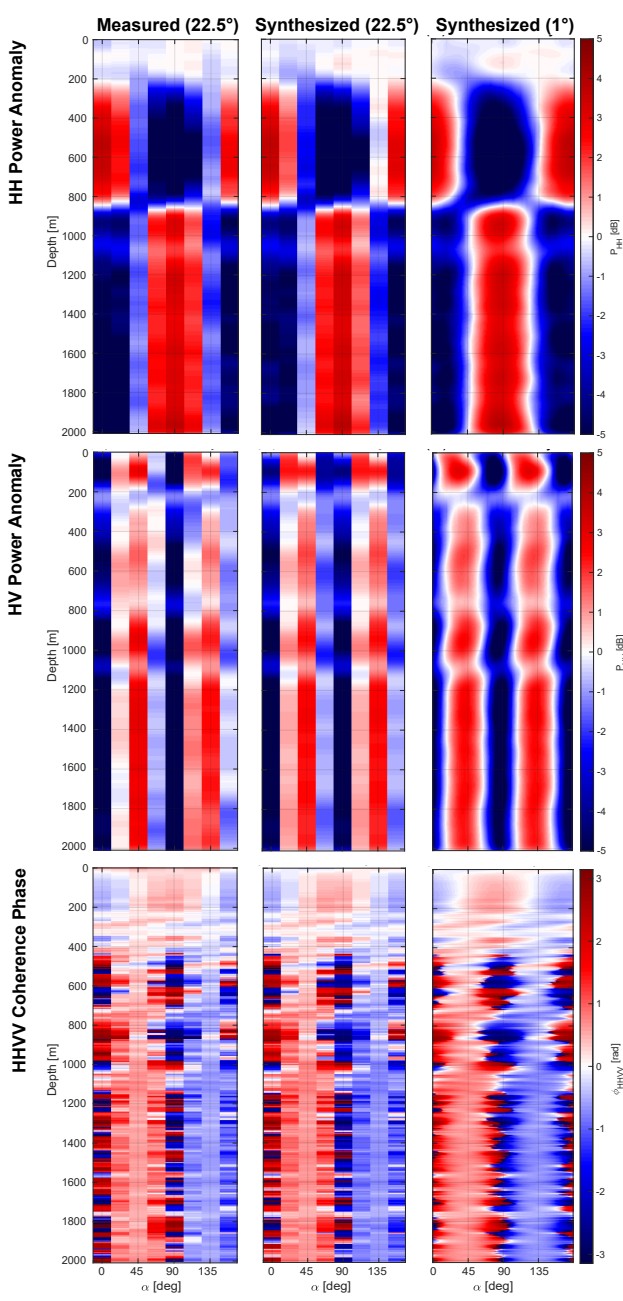

**Figure E1.** Comparison between collected and synthesized ApRES data at EDML site. (left column) collected ApRES data (22.5° azimuthal spacing). (middle column) synthesized ApRES data (22.5° azimuthal spacing). (right column) synthesized ApRES data (1° azimuthal spacing).





## Appendix F: Correlation between HH power anomaly ($P_{HH}$) nodes and anisotropic reflection ratio (r)

Here, we quantify the angular distance of co-polarization nodes ($AD$) as a function of the anisotropic reflection ratio ($r$). This
defaults to a two-dimensional minimization problem in $z$ and $\theta$ in the power anomaly $P_{HH}$. A co-polarization node in Eq. (C6)
requires

$$\cos(2zk_y) = -1. \tag{F1}$$

The remaining quadratic equation has two solutions corresponding to the two co-polarization nodes:

$$\theta_{node1} = \tan^{-1}(\frac{1}{\sqrt{r}} + \theta), \tag{F2}$$

$$\theta_{node2} = \tan^{-1}(\frac{1}{\sqrt{r}} - \theta). \tag{F3}$$

The angular distance between these nodes then results in

$$AD = |\theta_{node2} - \theta_{node1}| = 2\tan^{-1}(\frac{1}{\sqrt{r}}), \tag{F4}$$

which can be re-arranged for the reflection ratio as:

$$r = \frac{1}{\tan^2(\frac{AD}{2})}. \tag{F5}$$

*Author contributions.*  RE lead the code development and writing of the manuscript. RD, CM, and OE designed the study outline. RM, CR,
HC led the quad-polarimetric acquisition scheme and data collection at Dome C. JC, OZ, and AH lead data acquisition at EDML. All authors
contributed to the writing of the final manuscript.

*Competing interests.*  OE is Co-Editor in Chief and RD is Editor of The Cryosphere.

*Acknowledgements.*  Acknowledgements: RE and RD were supported by a DFG Emmy Noether grant DR 822/3-1. This publication was
also generated in the frame of Beyond EPICA. The project has received funding from the European Union's Horizon 2020 research and
innovation programme under grant agreement No. 815384 (Oldest Ice Core) and 730258 (CSA). It is supported by national partners and
funding agencies in Belgium, Denmark, France, Germany, Italy, Norway, Sweden, Switzerland, The Netherlands and the United Kingdom.
The Dome C measurements were made possible by the logistic provided by IPEV (prog. 902) and PNRA. We thank Luca Vittuari (University
of Bologna, Italy) for the positioning of the stakes. The opinions expressed and arguments employed herein do not necessarily reflect the
official views of the European Union funding agency or other national funding bodies. This is Beyond EPICA publication number XX. We
thank the AWI logistics personnel for support of the work at Kohnen.



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
