# Peer review of "Polarimetric radar reveals the spatial distribution of ice fabric at domes in East Antarctica"

_The Cryosphere, 2020_

## Referee Comment (RC1)

**Review to Ershadi et al. 2021**

**Summary**

Ershadi et al. 2021 introduce new technical developments within polarimetric radar-sounding to better estimate ice fabric. The novel aspect is an inversion/automation framework that uses radar reflection properties as an additional constraint to recover vertical fabric information, and hence the full second-order fabric orientation tensor (under the assumption the near-surface is isotropic). The inversion framework is validated via comparison with ice-core fabric data at EDML and EDC. In addition to the technical advances, the authors explore spatial variation in ice fabric around the core sites and relate to the ice-dynamic context.

The recovery of the full second-order fabric orientation tensor is a highly significant step forward on current polarimetric radar-sounding methods. It is also very useful for future studies that could use the method to parameterize anisotropic ice-flow models. The paper is well-written and referenced, with attractive figures, and, whilst being quite technical, is well-suited to TC.

Despite my overall positive impression, there are a number of things that should ideally be addressed to improve the paper - see specific comments below. The first of these comments is critical (and is the only reason why I recommended major corrections), as I think there is an inconsistency regarding the definition of `anisotropic scattering' that could require a re-run of the model inversions. My hope/intuition is that any differences in overall results should be small, and my suggested change may result in improved performance of the inversion, as sensitivity should be reduced.

Best regards to the authors – I enjoyed reviewing their paper.

Tom Jordan, Plymouth Marine Laboratory

1. **Definition of anisotropic scattering parameter (E-field amplitude versus intensity/power)**

A key thing to fix within the MS is how the anisotropic scattering parameter (currently called r) is defined. In particular, I believe there is an inconsistency between the reflection matrix, eq. (11) (where r should be defined in terms of the ratio of the electric-field reflection amplitudes/diagonal matrix elements – e.g. Fujita at et al. 2006), and eq. (12) (where r represents an intensity reflection ratio).

From the E-field definition, and combining with the small-permittivity form of the Fresnel equations (Paren 1981), it follows that *r should be linear with respect to the vertical eigenvalue contrast*, following:

r=gamma_y/gamma_x=(lamda_2,i-lamda_2,i+1)/ (lamda_1,i-lamda_1,i+1),.

However, in eq. (12) of the MS , r=gamma_y/gamma_x, is defined to be the square of the above  - i.e. gamma_y and gamma_x are taken to represent intensity reflection coefficients, and hence *r is assumed to be quadratic with respect to the vertical eigenvalue contrast*. This results in the fabric eigenvalues having a different scaling relationship with the forward model than they should.

In terms of presentation in the MS this is easy to fix, and I'd recommend:

> i) Being explicit that gamma_x and gamma_y represent E-field amplitude coefficients (e.g. L 154), which in general can be complex and/or negative.

ii) Define eq. (12) as R=|r|^2 =(gamma_y/gamma_x)^2=(lamda_2,i+1-lamda_2,i)^2/ (lamda_1,i+1-lamda_1,i)^2, as an `intensity reflection ratio' (rho is another option as I can see R is used for rotation). By convention, most optics fields use lower case for E-field reflection amplitudes, with the upper case for intensity, so I think this the best solution here. Similar, for eq. (29).

iii) Continue to use R within the rest of the MS for the anisotropic scattering parameter (using [R]_dB notation could also be useful to distinguish dB from the linear R form).

In terms of the impact on the results, I am less certain. If it is not just a typo/lack of clarification then, due to the linear rather than quadratic scaling, there will probably be some differences in the results on the numerical inversion, and the results should be re-run. It also follows from the linearized version of eq. (12), that r can now be negative (due to `reflection polarity'), so the impact of this may need to be considered within the inversion. The inversion framework appears adaptable to this change, so I hope this ends up with similar end-results that match the core data.

**2. Fresnel assumption for anisotropic scattering**

Linked to the point above, in applying eq. (12) the author's make the assumption that the reflection coefficients can be described using the Fresnel equations. They therefore implicitly assume that the radar reflections originate from a single sharp contrast between two adjacent fabric layers within a radar range cell. It is my understanding that the Fujita scattering model does not make this assumption (it makes a less restrictive assumption that the reflection is specular, which can accommodate for multiple scatterers/vertical fabric contrasts within a range cell). Eq. (12) should therefore be introduced as an additional assumption to the standard scattering model framework. Note – I think for the high bandwidth resolution ApRES data this assumption likely to be less of an issue than lower bandwidth radar systems, where the observed reflection will be more likely to be aggregated over multiple interfaces.

**3. Motivation for inversion/automation framework**

The inversion/automation framework is a very useful step forward, but I think it could be better motivated for the reader who is less familiar with the sub-field. In my opinion (other than extracting the vertical eigenvalue), a key advantage is how the new method is able to extract fabric rotation within the ice column. Previous (not automated/'via inspection') methods only perform well for the case that the fabric is depth-invariant, so extending to extract rotation within the ice column is significant.

**4. Significance of coherence magnitude**

One of the key advantages to using coherence methods is that the coherence magnitude, |c_hhvv| gives an idea of uncertainty/phase error. It would therefore be useful if the authors could: first introduce the significance of |c_hhvv| in terms of phase error; second include depth-azimuth plots of |c_hhvv| along-side phi_hhvv. To preserve the figure labels, this could even be done as a grey-scale underlay, as is sometimes done in the inSAR literature. Low coherence magnitude can be an issue in applying polarimetric methods to other regions of the ice sheets (particularly where radar layering is less ordered), so this would be very useful for comparative purposes, and to justify what parts of the ice column phi_hhvv can/cannot be trusted within the inversion.

5. **Incorporation of uncertainty within inversion**

This relates to the point above, and within the inversion framework, I think there are opportunities to incorporate more information on measurement uncertainty. For example, $|c_{hhvv}|$ and its relationship with phase uncertainty could be used to weight different depth/azimuth bins in eq. (23). Additionally, the authors mention that there is a ±15° uncertainty for georeferencing the data (which seems higher than I would have expected!). In theory, this alignment-uncertainty could be propagated via the quad-multi pol basis change into the scattering model and inversion. As significant developments have already been made in this paper, I think its ok for the authors to flag these as desirables for future work if this ends up being highly time-intensive.

6. **Plotting of phase gradient sign/polarity as evidence for fabric orientation and rotation**

Is there a good reason why the scaled phase derivative is defined as being positive in the data analysis (Fig. 5 and 6), whereas in the synthetic model it can be positive or negative (Fig. 3)? The sign of the phase gradient is the easiest way for the reader to connect the data analysis to the fabric orientation, and azimuthal rotations within the ice column (as explained in Appendix B). As it stands, I don't think the reader will be able to easily identify that there is an azimuthal rotation in Fig. 6, therefore, I recommend plotting the phase gradient with sign/polarity rather than the magnitude in the results figures. Also, I think it would make sense to better guide the reader about the `polarity depth-transition' being the marker of the fabric rotation within the ice column (Fig. 6).

7. **Cross-check with `vertical variability' of ice-core eigenvalue data**

It would be interesting to compare the ratio of the vertical variability of the lambda_1 and lambda_2 ice-core eigenvalues, with the inferred anisotropic scatting ratio. Whilst the vertical resolution of the ice cores will be different from the eigenvalue contrast related to a radar reflection, I would nevertheless expect them to be correlated – e.g. when $|r|$ is relatively high, I would expect the vertical variability of lambda_1 to be significantly different to lambda_2. This would be very interesting to comment on, and from looking at Fig. 5 and 6, I think there is likely to be different vertical variability of lambda_1 and lambda_2 present. Related to this – I wonder if the authors know of any dynamic significance of there being higher/lower variability in lambda_1 versus lamda_2? As the vertical r profiles are a novel feature of this paper, this would be good to comment on.

8. **General significance of recovering the vertical eigenvalue for ice-flow modeling**

The recovery of the full second-order tensor is highly significant for parameterizing ice-flow modeling with radar. To maximize the impact of the paper, I think it would be nice to add a brief section to the discussion on why this is the case. In particular: measurement of lambda_3 will give an indication of how the fabric ice becomes softer to shearing with the ice column. Some relevant anisotropic ice-flow references have been included, so these could be incorporated in this discussion here.

9. **Alignment with ice flow (versus compression) & glaciological interpretation of fabric rotation**

It is my understanding that rheological theory predicts that v_2 should be aligned with the lateral compression axis. For an extensional flow, the fabric is therefore is predicted to be perpendicular to the ice flow direction. I was therefore slightly surprised that v_2 tends to be aligned with the flow in the near-surface at EDML. Is it because near-surface compression acts along flow here? This relates to: L390-393, as are maximum and minimum the wrong way around here? I originally assumed that v2

should be aligned with across-flow compression (minimum strain) and v1 with the along-flow extension (maximum strain). In summary: it may make more sense to plot against the lateral compression axis at aswell/instead of ice flow.

Related to this point, an azimuthal rotation of the fabric is inferred at ice depth ~ 200m at EDML. As far as I can see, the glaciological significance of this isn't discussed in any detail. It would be useful to comment on this within the discussion, particularly in relation to the understanding of ice-flow history in the region (i.e. do we expect the lateral compression axis to have been different in the past?).

**10. Notation/terminology conventions**

I appreciate that notation can be subjective, but nevertheless, here are some suggestions that I have:

*Theta/alpha angle convention*. I think this is the other way round than in some other papers. For example, Matsuoka et al. 2012, Jordan et al. 2019, use alpha for the angle between the polarization planes and the eigenvectors, and theta for the geographic angle, whereas it is the other way round here). I don't think this matters, but this distinction with the other studies should be made clear.

*E-field notation.* It is also unusual, I think, to use lower case **e** for electric field. I'd recommend **E** as **e** is typically used for basis vectors in polarimetry studies.

*HV notation*. I was slightly confused by this, as it appears that H and V are used to describe measurements in both a fixed basis (i.e. the fixed geographic system where the quad pol data is measured), and a rotating basis (the coordinate system that phi_HHVV is analyzed in to extract the fabric). In Jordan 2019, 2020 H and V are reserved for the rotating-basis (i.e. they are a function of theta). If the authors want to make their description of phi_HHVV equivalent, then this distinction needs to be made clear – e.g. use H_0, V_0 for the quad-pol/fixed basis.

*Scattering matrix* – by convention the upper right element is normally HV (or 12, xy etc), and the lower left is VH.

*Power anomalies* – as Pxx , eq .(19) is a `power anomaly', I favour `Delta P' notation (I think Matsuoka et al. 2012, sets a good convention to follow for polarimetric power standards).

**Minor comments/technical corrections**

Should Eigenvalue and Eigenvectors be lower-case?

**Abstract**

It may be helpful that state you can `recover both horizontal and vertical anisotropy' with the inversion method, whereas previous radar methods can only recover horizontal. If it were me, I would probably state the key limitation of the method in the abstract and conclusion (the fact that isotropic ice needs to be present in the near-surface is a pretty strict requirement).

L4  It is important to note that in the inversion framework, the anisotropic reflection information is being `added' to ice birefringence - e.g. state as: ` ..approach within radar polarimetry, that combines ice birefringence and anisotropic reflection...'

L4 – It is helpful to be specific and put `complete second-order orientation tensor' (due to the fourth-order representation that can be used in seismics).

L11 – I think `that ice fabric horizontal distribution' needs re-wording

**Introduction**.

L31 – As this follows the section on optical microscopy, I think It is important to better-qualify that radar is observing a fundamentally different dielectric anisotropy of ice, which is the combined effect of the ice crystal birefringence and the crystal orientation fabric. Due to this  distinction, I'd say that radar is employing `similar' rather than the `same' principles to optical methods to recover fabric (i.e. it is based on measuring a bulk anisotropy rather than an intrinsic).

L39 –Is there a typical value for `high resolution' to add here?

L49 – Li et al. 2018 is another good reference for applications to ice domes: https://tc.copernicus.org/articles/12/2689/2018/tc-12-2689-2018.pdf

L51 – To the best of my knowledge, Robert et al., 1993; Joughin et al., 1999 are not examples of ice-penetrating radar polarimetry/fabric estimation within ice streams (at least, not in the same sense as the domes, divides, and rises references). Maybe they should be added elsewhere?

L54 – It would be helpful to guide the reader here. e.g. `forward modeling framework which relates the vertical distribution of ice fabric to the polarimetric radar signal'

Table 1 - This table is very useful to include! I think `Dielectric permittivity matrix' should be more precisely defined as `principal dielectric tensor'.

**Section 2**

L 67 the ice flow

L70/71 – I assume ` largest principal strain' and `maximum strain' will be the same?  If so, it would be helpful to use the same term throughout.

**Section 3**

3.1. Within this section, I would briefly describe how the 1,2,3 axes are typically assumed to orientated relative to the vertical.

L 78 –I'd  check the wording of this sentence

L 80 – It would be helpful here to be specific that you are describing the second-order orientation tensor representation

L 124 birefringence  >birefringent

L185 -I assume the derivative was applied to the real and imaginary components of the complex coherence?  This should be made explicit.

**Section 5**

L331 I'd replaced `dielectric orientation tensor' with `second-order fabric orientation tensor ' (strictly the dielectric tensor is a separate object).

L339 – `*However, more systematic investigation is warranted if this also holds when the ice fabric is not aligned vertically.'* I think the basis transform/polarization synthesis should hold irrespective of the vertical alignment, so multi-pol should always map to quad-pol for the case of no georeferencing error). The respective SNR could be different though.

**Appendices**

B - I think it would be helpful to guide the readers why a `triangular space' arises in Fig B1 (due to the constraints lambda_1+lambda_2+lambda_3=1 and lambda_3>lambda_2>lambda_1).

C- I think z_1 should be z in eq. C5?

D – This derivation is nice to include. I think if you were to remind the readers how the k components relate to permittivity or refractive index and phase velocity, then it would help guide the reader.

E  - I like the quad/multi-pol comparison! I think comparing the coherence magnitude and power-SNR are the most interesting thing to look at here, as these are what will differ between the methods. The inclusion of cross-mode (HV, VH) terms, which generally have lower amplitudes than HH, VV is relevant.

F – If you agree with my change to eq. (12), then sqrt(r )should be replaced by r , and r with R.

**Figures**

Fig. 1 - If it is easy to do, then it would be helpful to include a velocity scalebar.

---

## Author Comment (AC1)

**Response to the review of Ershadi et al. 2021 "Polarimetric radar reveals the spatial distribution of ice fabric at domes in East Antarctica."**

**[Manuscript # tc-2020-370]**

**RC1. Reviewed by Tom Jordan (07.02.2021).**
**RC2. Reviewed by Anonymous Reviewer (26.04.2021).**

Dear Editor and Reviewers,

Thank you for your valuable feedback on our paper. We perceived the concerns raised as constructive and helpful and have implemented many changes accordingly. Here, we reply to both reviewers (RC1 & RC2) point-by-point. The reviewer's comments (RC) are marked in blue, followed by our response (AC) in black. Numbering of equations, figures, and line numbers refer to the original version of the MS unless noted otherwise.

Some of the reviewer's comments diverged in the sense that the paper should preferably be shorter (reviewer 2) but also offers some avenues for further exploration (reviewer 1). Reconciling both views entirely is impossible, but we accommodated the remarks by shortening parts of the methods and expanding parts of the discussions. Most critically, we fixed a mistake pointed out by reviewer 1 related to the Fresnel reflection coefficients. Fortunately, this has had no follow-up impacts on our methodological approach and leaves the results & interpretations & conclusions unchanged.

We would like to thank both reviewers for their time and the constructive comments. We are particularly thankful to Tom Jordan (reviewer 1) for a detailed review pointing out a mistake that would have been missed otherwise.

**On behalf of all co-authors,**

**M. Reza Ershadi (University of Tübingen)**

**AC to RC1 (TOM JORDAN):**

**Summary**
Ershadi et al. 2021 introduce new technical developments within polarimetric radar-sounding to better estimate ice fabric. The novel aspect is an inversion/automation framework that uses radar reflection properties as an additional constraint to recover vertical fabric information, and hence the full second-order fabric orientation tensor (under the assumption the near-surface is isotropic). The inversion framework is validated via comparison with ice-core fabric data at EDML and EDC. In addition to the technical advances, the authors explore spatial variation in ice fabric around the core sites and relate to the ice-dynamic context.

The recovery of the full second-order fabric orientation tensor is a highly significant step forward on current polarimetric radar-sounding methods. It is also very useful for future studies that could use the method to parameterize anisotropic ice-flow models. The paper is well-written and referenced, with attractive figures, and, whilst being quite technical, is well-suited to TC.

Despite my overall positive impression, there are a number of things that should ideally be addressed to improve the paper - see specific comments below. The first of these comments is critical (and is the only reason why I recommended major corrections), as I think there is an inconsistency regarding the definition of `anisotropic scattering' that could require a re-run of the model inversions. My hope/intuition is that any differences in overall results should be small, and my suggested change may result in improved performance of the inversion, as sensitivity should be reduced.

Best regards to the authors – I enjoyed reviewing their paper.
Tom Jordan, Plymouth Marine Laboratory

**AC.** Thank you for the positive feedback. It was a very constructive review, and the major correction you suggested in the first comment helped us avoid having a mistake in the MS. We are also grateful for your suggestions in several areas where this paper can be expanded. We implemented some of them and decided to postpone others to future papers (e.g., a more

detailed analysis about azimuthal reconstruction vs. azimuthal measurements) given that this paper is already quite long (cf. reviewer 2).
* * *
**1-Definition of anisotropic scattering parameter (E-field amplitude versus intensity/power)**

A key thing to fix within the MS is how the anisotropic scattering parameter (currently called r) is defined. In particular, I believe there is an inconsistency between the reflection matrix, eq. (11) (where r should be defined in terms of the ratio of the electric-field reflection amplitudes/diagonal matrix elements – e.g. Fujita at et al. 2006), and eq. (12) (where r represents an intensity reflection ratio).

From the E-field definition, and combining with the small-permittivity form of the Fresnel equations (Paren 1981), it follows that r should be linear with respect to the vertical eigenvalue contrast, following:

r=gamma_y/gamma_x=(lamda_2,i-lamda_2,i+1)/ (lamda_1,i-lamda_1,i+1),.

However, in eq. (12) of the MS , r=gamma_y/gamma_x, is defined to be the square of the above - i.e. gamma_y and gamma_x are taken to represent intensity reflection coefficients, and hence r is assumed to be quadratic with respect to the vertical eigenvalue contrast. This results in the fabric eigenvalues having a different scaling relationship with the forward model than they should.

In terms of presentation in the MS this is easy to fix, and I'd recommend:

   i) Being explicit that gamma_x and gamma_y represent E-field amplitude coefficients (e.g. L 154), which in general can be complex and/or negative.
   ii) Define eq. (12) as R=|r|^2 =(gamma_y/gamma_x)^2=(lamda_2,i+1-lamda_2,i)^2/ (lamda_1,i+1-lamda_1,i)^2, as an `intensity reflection ratio' (rho is another option as I can see R is used for rotation). By convention, most optics fields use lower case for E-field reflection amplitudes, with the upper case for intensity, so I think this the best solution here. Similar, for eq. (29).
   iii) Continue to use R within the rest of the MS for the anisotropic scattering parameter (using [R]_dB notation could also be useful to distinguish dB from the linear R form).

In terms of the impact on the results, I am less certain. If it is not just a typo/lack of clarification then, due to the linear rather than quadratic scaling, there will probably be some differences in the results on the numerical inversion, and the results should be re-run. It also follows from the linearized version of eq. (12), that r can now be negative (due to `reflection polarity'), so the impact of this may need to be considered within the inversion. The inversion framework appears adaptable to this change, so I hope this ends up with similar end-results that match the core data.

**AC.** We agree with your finding. Thanks for pointing this out. We have corrected Eq. (12), which has (fortunately) not had any follow-up consequences for the eigenvalue reconstruction. On the contrary, the reconstruction is now more stable than before. The inversion results remained the same as the parameterization of "r" is only used in the eigenvalue reconstruction. In the reorganization of the paper, the correct description of the reflection coefficient is now in Eq. (19) of the revised MS. We did not fully follow your suggestions to change the notations. In our view, it is clearer to add a ± square root to Eq. (12) and keep it as "r," which allows us to keep using "R" for the rotation matrix. Changes are accommodated in sections 3.3 and 3.6. Also, the result figures (Fig 5,6,7) are updated.
* * *
**2-Fresnel assumption for anisotropic scattering**

Linked to the point above, in applying eq. (12) the author's make the assumption that the reflection coefficients can be described using the Fresnel equations. They therefore implicitly assume that the radar reflections originate from a single sharp contrast between two adjacent fabric layers within a radar range cell. It is my understanding that the Fujita scattering model does not make this assumption (it makes a less restrictive assumption that the reflection is specular, which can accommodate for multiple scatterers/vertical fabric contrasts within a range cell). Eq. (12) should therefore be introduced as an additional assumption to the standard scattering model framework. Note – I think for the high bandwidth resolution ApRES data this assumption likely to be less of an issue than lower bandwidth radar systems, where the observed reflection will be more likely to be aggregated over multiple interfaces.

**AC.** Agreed. We mentioned this assumption in the first paragraph of section 3.6 of the revised MS.
* * *
**3-Motivation for inversion/automation framework**

The inversion/automation framework is a very useful step forward, but I think it could be better motivated for the reader who is less familiar with the sub-field. In my opinion (other than extracting the vertical eigenvalue), a key advantage is how the new method is able to extract fabric rotation within the ice column. Previous (not automated/'via inspection') methods only perform well for the case that the fabric is depth-invariant, so extending to extract rotation within the ice column is significant.

**AC.** We added these points to the abstract, introduction, and discussion (Sect. 5.1) to better motivate the revised MS.
* * *
One of the key advantages to using coherence methods is that the coherence magnitude, |c_hhvv| gives an idea of uncertainty/phase error. It would therefore be useful if the authors could: first introduce the significance of |c_hhvv| in terms of phase error; second include depth-azimuth plots of |c_hhvv| along-side phi_hhvv. To preserve the figure labels, this could even be done as a grey-scale underlay, as is sometimes done in the inSAR literature. Low coherence magnitude can be an issue in applying polarimetric methods to other regions of the ice sheets (particularly where radar layering is less ordered), so this would be very useful for comparative purposes, and to justify what parts of the ice column phi_hhvv can/cannot be trusted within the inversion.

**AC.** Agreed. We state the usefulness of the coherence magnitude as a measure for the phase error in the revised MS (Sect. 3.3) and outsource further details to Jordan et al (2019 and 2020). We believe including the coherence magnitude plot as a gray-scale underlay will make the results figure more confusing for the reader rather than providing necessary information.
* * *
**5-Incorporation of uncertainty within inversion**
This relates to the point above, and within the inversion framework, I think there are opportunities to incorporate more information on measurement uncertainty. For example, |c_hhvv| and its relationship with phase uncertainty could be used to weight different depth/azimuth bins in eq. (23).

**AC.** This is a good idea, and this should be applied if we extend the automatic retrieval algorithm to larger depths with less coherence. In this paper, we have decided to limit ourselves to 2000 m where the coherence magnitude is reliable (mostly larger than 0.4 as suggested by Jordan et al. 2019). A weighing of eq. 23 with coherence values does hence not substantially change the optimization done here. We consider that the coherence magnitude and phase error concepts are well explained in Jordan et al. (2019), so we would like to avoid duplication.
* * *
Additionally, the authors mention that there is a ±15° uncertainty for georeferencing the data (which seems higher than I would have expected!). In theory, this alignment-uncertainty could be propagated via the quad-multi pol basis change into the scattering model and inversion. As significant developments have already been made in this paper, I think its ok for the authors to flag these as desirables for future work if this ends up being highly time-intensive.

**AC.** The ±15° error in georeferencing is the roughly estimated error from using a magnetic compass for aligning both antennas (which could be smaller indeed, but we try staying on the safe side). From our understanding propagating these uncertainties into the model would lead to a bulk change in the inferred ice-fabric direction. Considering a more rigorous treatment (e.g. a misalignment of only one antenna) is more complicated, but we will flag them as desirable for future work as suggested.
* * *
**6-Plotting of phase gradient sign/polarity as evidence for fabric orientation and rotation**
Is there a good reason why the scaled phase derivative is defined as being positive in the data analysis (Fig. 5 and 6), whereas in the synthetic model it can be positive or negative (Fig. 3)? The sign of the phase gradient is the easiest way for the reader to connect the data analysis to the fabric orientation, and azimuthal rotations within the ice column (as explained in Appendix B). As it stands, I don't think the reader will be able to easily identify that there is an azimuthal rotation in Fig. 6, therefore, I recommend plotting the phase gradient with sign/polarity rather than the magnitude in the results figures. Also, I think it would make sense to better guide the reader about the `polarity depth-transition' being the marker of the fabric rotation within the ice column (Fig. 6).

**AC.** In the MS, we have mentioned that the scaled phase derivative is anti-symmetric, where the positive side is in the direction of v2, and the negative side is in the direction of v1. We showed both positive and negative parts of the scaled phase derivative in the synthesized model to exhibit this. We believe that the scaled phase derivate plots are relatively complex for the readers. Therefore, for the rest of the MS, for simplicity and avoiding confusion, we masked the negative part of the plots so the readers can immediately recognize the direction of v2. We believe that masking the negative part of the scaled phased derivative does not cause any confusion or losing information regarding the orientation of the fabric and the value of horizontal anisotropy.
* * *
**7-Cross-check with `vertical variability' of ice-core eigenvalue data**
It would be interesting to compare the ratio of the vertical variability of the lambda_1 and lambda_2 ice-core eigenvalues, with the inferred anisotropic scatting ratio. Whilst the vertical resolution of the ice cores will be different from the eigenvalue contrast related to a radar reflection, I would nevertheless expect them to be correlated – e.g. when |r| is relatively high, I would expect the vertical variability of lambda_1 to be significantly different to lambda_2. This would be very interesting to comment on, and from looking at Fig. 5 and 6, I think there is likely to be different vertical variability of lambda_1 and lambda_2 present. Related to this – I wonder if the authors know of any dynamic significance of there being higher/lower variability in lambda_1 versus

lamda_2? As the vertical r profiles are a novel feature of this paper, this would be good to comment on.

**AC.** Drews et al. (2012) analyzed the variation of the eigenvalues with depth for the EDML ice core and found that a change in radar backscatter anisotropy coincides with a change in the variability of the eigenvalues. They potentially linked them to changes in the impurity content coming along with glacial-interglacial cycles. The eigenvalues at Dome C are coarsely resolved and hence do not provide a good measure for the reflection coefficients.
* * *
**8-General significance of recovering the vertical eigenvalue for ice-flow modeling**
The recovery of the full second-order tensor is highly significant for parameterizing ice-flow modeling with radar. To maximize the impact of the paper, I think it would be nice to add a brief section to the discussion on why this is the case. In particular: measurement of lambda_3 will give an indication of how the fabric ice becomes softer to shearing with the ice column. Some relevant anisotropic ice-flow references have been included, so these could be incorporated in this discussion here.

**AC.** This has been added to the introduction and discussion (Sect. 5.1) of the revised MS.
* * *
**9-Alignment with ice flow (versus compression) & glaciological interpretation of fabric rotation**
It is my understanding that rheological theory predicts that v_2 should be aligned with the lateral compression axis. For an extensional flow, the fabric is therefore is predicted to be perpendicular to the ice flow direction. I was therefore slightly surprised that v_2 tends to be aligned with the flow in the near-surface at EDML. Is it because near-surface compression acts along flow here? This relates to: L390-393, as are maximum and minimum the wrong way around here? I originally assumed that v2 should be aligned with across-flow compression (minimum strain) and v1 with the along-flow extension (maximum strain). In summary: it may make more sense to plot against the lateral compression axis at aswell/instead of ice flow.

**AC.** The surface velocity and strain rate data are copied directly from other publications. They both show along-flow compression (minimum rate-short axis of the ellipse) and across-flow extension (maximum rate-long axis of the ellipse). We agree with the reviewer that we tend to imagine ice extending in the direction of flow, but this is not necessarily true. In fact, it does not apply to the two examples presented in our paper where ice is compressing in the direction of flow. We will make this point clearer in the discussion around the interpretation of the fabric direction.
* * *
Related to this point, an azimuthal rotation of the fabric is inferred at ice depth ~ 200m at EDML. As far as I can see, the glaciological significance of this isn't discussed in any detail. It would be useful to comment on this within the discussion, particularly in relation to the understanding of ice-flow history in the region (i.e. do we expect the lateral compression axis to have been different in the past?).

**AC.** This is indeed an excellent point. We have also inferred an azimuthal rotation for the top 200 m at EDC. Generally speaking, the fabric orientation estimated from the top part of the ice column, which can be assumed isotropic, cannot be reliable. When we are talking about the isotropic part, the orientation becomes meaningless. If you check the scaled phase derivative for the top 200 m at EDC, the value of inferred horizontal anisotropy in that part can be more or less the same regardless of the presumed orientation. This is different at EDML, where the orientation for the top 200 m is reliable. We added a short paragraph explaining this in the discussion of the revised MS (Sect. 5.2).
* * *
**10-Notation/terminology conventions**
I appreciate that notation can be subjective, but nevertheless, here are some suggestions that I have:

**Theta/alpha angle convention**. I think this is the other way round than in some other papers. For example, Matsuoka et al. 2012, Jordan et al. 2019, use alpha for the angle between the polarization planes and the eigenvectors, and theta for the geographic angle, whereas it is the other way round here). I don't think this matters, but this distinction with the other studies should be made clear.

**AC.** We chose theta to be consistent with Fujita et al. (2006). Alpha is indeed defined differently, but it is such a generic variable that we hope the reader does not have a priori understanding of what it should be since alpha is an established symbol for azimuth.
* * *
**E-field notation**. It is also unusual, I think, to use lower case e for electric field. I'd recommend E as e is typically used for basis vectors in polarimetry studies.

**AC.** Agreed and corrected.
* * *
**HV notation**. I was slightly confused by this, as it appears that H and V are used to describe measurements in both a fixed basis

(i.e. the fixed geographic system where the quad pol data is measured), and a rotating basis (the coordinate system that phi_HHVV is analyzed in to extract the fabric). In Jordan 2019, 2020 H and V are reserved for the rotating-basis (i.e. they are a function of theta). If the authors want to make their description of phi_HHVV equivalent, then this distinction needs to be made clear – e.g. use H_0, V_0 for the quad-pol/fixed basis.

**AC.** In this study, we only refer to quad-pole measurements, which means all the four measurements (HH, VV, VH, and HV) are acquired at the same alpha (and theta), meaning TR aerial line was not rotating. We believe that adding H_0 and V_0 is confusing. Therefore, the description of phi_HHVV in this study is equivalent to Jordan et al. (2019 and 2020).
* * *
**Scattering matrix** – by convention the upper right element is normally HV (or 12, xy etc), and the lower left is VH.

**AC.** We agree that by convention HV is the top right element of the scattering matrix, and VH is the bottom left. We decided to change that in this study, and here is the reason. In the equations below, the left-hand side is the received signal in the H and V direction, wherein (A) the signal is transmitted in the H direction, and in B, the signal is transmitted in the V direction.

$$(A): \begin{bmatrix} a: Received\ in\ H\ direction \\ c: Received\ in\ V\ direction \end{bmatrix} = \begin{bmatrix} a & b \\ c & d \end{bmatrix} * \begin{bmatrix} 1: Transmitted\ in\ H\ direction \\ 0 \end{bmatrix}.$$

$$(B): \begin{bmatrix} b: Received\ in\ H\ direction \\ d: Received\ in\ V\ direction \end{bmatrix} = \begin{bmatrix} a & b \\ c & d \end{bmatrix} * \begin{bmatrix} 0 \\ 1: Transmitted\ in\ V\ direction \end{bmatrix}.$$

As a result, the components of the scattering matrix are named as follows:
a is sHH (transmitted in H, received in H),
c is sHV (transmitted in H, received in V),
b is sVH (transmitted in V, received in H),
d is sVV (transmitted in V, received in V).
* * *
**Power anomalies** – as Pxx , eq .(19) is a `power anomaly', I favour `Delta P' notation (I think Matsuoka et al. 2012, sets a good convention to follow for polarimetric power standards).

**AC.** Agreed and corrected.
* * *
**Minor comments/technical corrections**
Should Eigenvalue and Eigenvectors be lower-case?

**AC.** Not sure but we chose the lower-case now.
* * *
**Abstract**
It may be helpful that state you can `recover both horizontal and vertical anisotropy' with the inversion method, whereas previous radar methods can only recover horizontal.

**AC.** Agreed and added.
* * *
If it were me, I would probably state the key limitation of the method in the abstract and conclusion (the fact that isotropic ice needs to be present in the near-surface is a pretty strict requirement).

**AC.** We believe this is not a real limitation and is not worth to be mentioned in the abstract. In the revised MS we explained this in detail in section 3.6.
* * *
**L4** – It is important to note that in the inversion framework, the anisotropic reflection information is being `added' to ice birefringence - e.g. state as: ` ..approach within radar polarimetry, that combines ice birefringence and anisotropic reflection…'

**AC.** Agreed and corrected.
* * *
**L4** – It is helpful to be specific and put `complete second-order orientation tensor' (due to the fourth-order representation that can be used in seismics).

**AC.** Agreed and corrected.
* * *
**L11** – I think `that ice fabric horizontal distribution' needs re-wording

**AC.** Agreed and corrected.
* * *
**Introduction**
**L31** – As this follows the section on optical microscopy, I think It is important to better-qualify that radar is observing a fundamentally different dielectric anisotropy of ice, which is the combined effect of the ice crystal birefringence and the crystal orientation fabric. Due to this distinction, I'd say that radar is employing `similar' rather than the `same' principles to optical methods to recover fabric (i.e. it is based on measuring a bulk anisotropy rather than an intrinsic).

**AC.** Agreed and added.
* * *
**L39** –Is there a typical value for `high resolution' to add here?

**AC.** Added to the revised MS as sub-cm scale.
* * *
**L49** – Li et al. 2018 is another good reference for applications to ice domes: https://tc.copernicus.org/articles/12/2689/2018/tc-12-2689-2018.pdf

**AC.** Thank you for suggesting this. Added to the revised MS.
* * *
**L51** – To the best of my knowledge, Robert et al., 1993; Joughin et al., 1999 are not examples of ice-penetrating radar polarimetry/fabric estimation within ice streams (at least, not in the same sense as the domes, divides, and rises references). Maybe they should be added elsewhere?

**AC.** Agreed and corrected.
* * *
**L54** – It would be helpful to guide the reader here. e.g. `forward modeling framework which relates the vertical distribution of ice fabric to the polarimetric radar signal'

**AC.** Agreed and added.
* * *
**Table 1** - This table is very useful to include! I think `Dielectric permittivity matrix' should be more precisely defined as `principal dielectric tensor'.

**AC.** Agreed and corrected.
* * *
**Section 2**
**L 67** –  the ice flow

**AC.** Agreed and corrected.
* * *
**L70/71** – I assume ` largest principal strain' and `maximum strain' will be the same? If so, it would be helpful to use the same term throughout.

**AC.** Agreed and corrected.
* * *
**Section 3**
**3.1.** Within this section, I would briefly describe how the 1,2,3 axes are typically assumed to orientated relative to the vertical.

**AC.** In Section 3.1. we are trying only to introduce the axes. The specific condition for this study and our method (lambda3 in vertical and lambda1&2 in horizontal) has been explained in section 3.3.
* * *
**L 78** –I'd check the wording of this sentence

**AC.** Agreed and corrected.
* * *
**L 80** – It would be helpful here to be specific that you are describing the second-order orientation tensor representation

**AC.** Agreed and added.
* * *
**L 124** – birefringence >birefringent

**AC.** Agreed and corrected.
* * *
**L185** – I assume the derivative was applied to the real and imaginary components of the complex coherence? This should be

made explicit.

**AC.** Agreed and added.
* * *
**Section 5**
**L331** – I'd replaced `dielectric orientation tensor' with `second-order fabric orientation tensor '(strictly the dielectric tensor is a separate object).

**AC.** Agreed and corrected.
* * *
**L339** – `However, more systematic investigation is warranted if this also holds when the ice fabric is not aligned vertically.' I think the basis transform/polarization synthesis should hold irrespective of the vertical alignment, so multi-pol should always map to quad-pol for the case of no georeferencing error). The respective SNR could be different though.

**AC.** Agreed and corrected.
* * *
**Appendices**
**B** - I think it would be helpful to guide the readers why a `triangular space' arises in Fig B1 (due to the constraints lambda_1+lambda_2+lambda_3=1 and lambda_3>lambda_2>lambda_1).

**AC.** Agreed and added.
* * *
**C-** – I think z_1 should be z in eq. C5?

**AC.** Agreed and corrected.
* * *
**D** – This derivation is nice to include. I think if you were to remind the readers how the k components relate to permittivity or refractive index and phase velocity, then it would help guide the reader.

**AC.** We decided to shorten the method section. Therefore, the explanation of k component is moved to appendix B.
* * *
**E** – I like the quad/multi-pol comparison! I think comparing the coherence magnitude and power-SNR are the most interesting thing to look at here, as these are what will differ between the methods. The inclusion of cross-mode (HV, VH) terms, which generally have lower amplitudes than HH, VV is relevant.

**AC.** We agree with you. This is very interesting to see. But we decided not to include more details about this topic in this MS. We will leave this for future studies which will focus more on synthesizing quad-pole measurements.
* * *
**F** – If you agree with my change to eq. (12), then sqrt(r )should be replaced by r , and r with R.

**AC.** We kept "R" for the rotation matrix. Therefore no change is needed here.
* * *
**Figures**
**Fig. 1** - If it is easy to do, then it would be helpful to include a velocity scalebar.

**AC.** In this map, the yellow arrows show the surface velocity and its orientation. Each map has one arrow with the magnitude of the velocity written on top of it as a scale. We believe this is sufficient as a scale for the velocity.
* * *
**Authors' response to RC2 (ANONYMOUS):**

The paper by Reza Ershadi et al documents the utility of polarimetric radar and derived parameters such as power and phase to map ice-fabric, critical for ice flow observations and modeling. With my expertise in satellite radar polarimetry (and with a limited experience in ice fabric), my brief notes (for this round of review) below will focus more towards the pros and cons of this manuscript related to the polarimetric aspect of this study.

Overall, the authors have presented a novel observational approach, supported by strong modeling framework towards baseline retreivals of ice fabric parameters (orientation, anisotropy and their vertical variabiity) from the two sites in the Antarctic; which makes the paper **compelling** and **impactful** to be published in TC. There are some general comments though, which requires revision, focused towards paper structuring (conciseness) and interpretation, before which the paper can be accepted.

I **keep this round of review brief** owing to the reasons I mention below with the Paper Structuring and Conciseness. I am

willing to review the revised manuscript where I will include specific comments related to presentation quality etc.

**AC.** We appreciate your time and effort in this review. We will answer your comments in the best way we can.
* * *
**Paper Structuring**: In general, I find the paper (excluding the appendix) to be way too long, with a substantial focus given to the methods (which is OK for me), but diverts the attention away from the interesting results and discussions. By the time, I finished methods, I found 31 equations in the methods to be exhaustive, although the authors took time to have Table 1 with the list of symbols. Here is amy suggestion. Section 3.3 can be refined by referencing the methods, and shortening the section by explaining how radar polarimetry is used for this particular study. This will really help the readers to follow through the objectives and methods with a strong connect, and make the paper more concise.

**AC.** We appreciate your thoughts and suggestions on this. Unfortunately, there is never sufficient agreement about the optimal level of detail to be included in a technical paper. In fact, reviewer 1 is asking for more details to be included to make the study comprehensive. Our view is that this paper is introducing a new method to process polarimetric radar data. Therefore we have to describe the foundations of the field to explain our development within the context and clarity. Without the mathematical background, it will be difficult for the readers to follow the plausibility of our findings. Our target group is not only remote sensing or radioglaciology experts but also glaciologists from other fields which are less familiar with the polarimetric background. For those, we provide a sufficiently comprehensive and self-contained description of the matter in this manuscript. In order to accommodate your concerns, we shortened section 3.3 by moving some of the content to Appendix B.
* * *
**Results and Discussion:** At the EDC site, the authors clearly illustrate the co- polarization node (CPN) at 1100 m and at around 2000 m. Especially the HH and VV phase coherence shows strong phase shifts at these depths. Just curious, but why dont (or what is the reason), the derived anisotropy shows almost negligible change in measured or estimated eigen values, vectors and anisotropy at these depths. From my knowledge, when there is a 90 degree phase shift (most prominent at 1100 m), that change demonstrable of anisotropic scattering (as a function of change in ice grain size ororientation).

**AC.** Indeed you mentioned an excellent point here. The answer is the depth of the node (power anomaly), or the depth of the phase shift (coherence phase) is an integration of the horizontal anisotropy above that depth and not at that depth. In other words, the strong phase shift will occur no matter how strong or weak the anisotropy is. It will just occur at lower or higher depth. In summary, the occurrence of the strong phase shift at a specific depth does not mean the anisotropy at that depth must be necessarily stronger. Also, the fact that the horizontal anisotropy and eigenvalues that we inferred are very similar to the lab analysis confirms the absence of any abrupt change. In any case, your point has made us realize that we should caution our readers not familiarized with radar polarimetry by incorporating an explanation of this effect in the revised version of the MS.
* * *
I find this vertical homogeneity even more interesting, when we compare them to the displayed vertical heterogenity in fabric parameters from the EDML site. This is in slight contradiction to your statement in the discussion' **Multi-layer cases, however, are difficult to interpret, particularly if the ice-fabric orientation changes strongly (by several 10s of degrees) with depth**. Fortunately, **this does not appear to be the case for the data presented here**, so that the initial guess already results in a forward model that **adequately captures characteristic features in the data**'.

**AC.** Please note that the statement you quoted from the MS talks about the **fabric orientation** and not the **fabric anisotropy**. The fabric anisotropy is presented in subplot (j) in both figures (5 and 6), while fabric orientation is shown in subplot (k). The statement you quoted refers to subplot (k), wherein both sites exhibit a depth invariant fabric orientation meaning the fabric orientation is not changing with depth (except the top 200m, which is not problematic). Therefore, there is no contradiction in the mentioned sentence. To make this clearer, we referenced the parameters and the figures in the revised MS.
* * *
I think, since both sites indicate demonstrable differences in derived ice-fabric parameters(across depth) (based on Figure 5 and 6), that shows the applicability of the method usedin this study. But at the same time, the paper lacks explanation about why there are site-specific differences, which will be a neat discussion to include.

**AC.** We totally agree with the reviewer that it will be interesting to explain the significant

differences in the fabric between the two sites regarding flow conditions. We have hinted at this in different sections of the MS, but we will incorporate a short new paragraph in Section 5.2 highlight this point.

---

## Referee Report (RR1)

**Second review to Ershardi et al. 2021,**

The authors' have given extensive feedback to my review comments, and implemented changes where appropriate. I appreciate that the other reviewer found the MS to be overtly technical/equation dense. However - the MS is broadly in-line with other respected polarimetry/ice fabric papers (e.g. those by Matusoka and Fujita). Some new technical developments were made, hence justifying the level of detail of the presentation. Additionally, both the introduction and discussion are now improved, so I think general TC readers could enjoy those sections without needing to fully understand the whole paper. In summary - I think the re-submitted MS is considerably tighter than before, and I recommend publication subject to a few final comments.

Tom Jordan, Plymouth Marine Laboratory, 24/10/2021
* * *
**Feedback on specific points from last review**

1. *Anisotropic scattering parameter.* This has now been implemented by taking the positive square-route of the intensity ratio (eq. 19). I agree this is an improvement from before, as the scattering parameter is now linear w.r.t. the eigenvalue difference, and is therefore consistent with the matrix model scaling relationship. I am also happy to see that the suggested change did not have an adverse effect on the previous conclusions. *However, notation-wise I still think eq. (6) (where r is complex number) an eq. (19) (where r is a scalar) are inconsistent.* My recommendation would be to use |r| instead of r in eq. (19) (and all subsequent usage) and be clearer throughout the MS that it represents a magnitude.

*8. Significance of recovering the vertical eigenvalue for ice-flow modeling.* I think this could be made more explicit, as this is a key step forward for what the radar can provide to ice-sheet models (and it is considerably more useful than the `horizontal eigenvalue difference' in constraining rheology). Ice-sheet modelers' will be interested in this result, as the strength of the vertical eigenvalue is a strong control on how soft ice is to shearing within the ice column - e.g. see: Azuma 1996 - https://www.cambridge.org/core/journals/annals-of-glaciology/article/an-anisotropic-flow-law-for-icesheet-ice-and-its-implications/A9AC75ED14AD578C809CCF9752BF1D20

All other comments look well implemented (or justified) to me - good job!
* * *
**Additional comments**

**L68 –** Maybe `parameterization of an anisotropic flow law' works better than `further developments of…'?

**Fig. 5 caption –** it looks like some formatting errors are present

**L 522** `Since polarimetric radar is insensitive to the vertical component of ice fabric, it is only possible to estimate its horizontal anisotropy (Sect. 3.3).' I think the readers may find this line confusing as the paper obviously developed the inversion method to solve for all 3 eigenvalues. Maybe – add `… its horizontal anisotropy from the matrix model alone'

---

## Referee Report (RR2)

[revised manuscript text omitted]
 $\lambda1_1 \approx 0.33$ allowing to infer $\lambda2_1$ and $\lambda3_1$ from the estimated $\Delta\lambda_1$

$$\lambda2_1 = \Delta\lambda_1 + \lambda1_1,$$  (20)

$$\lambda3_1 = 1 - \lambda2_1 - \lambda1_1.$$  (21)

The eigenvalues for the surface can be estimated by iterating through Eqs. (20) and (21) and decreasing the value of $\lambda1_1$ by $1.0 \cdot 10^{-5}$ at each iteration until all the surface eigenvalues fulfill the requirements in Sect. 3.1. For deeper layers $i+1$, all three eigenvalues, can be reconstructed analytically by solving

$$\lambda1_{i+1} = \lambda1_i - \left(\frac{\Delta\lambda_i - \Delta\lambda_{i+1}}{r_i - 1}\right)$$  (22)

for $\lambda1_{i+1}$ and infer $\lambda2_{i+1}$ and $\lambda3_{i+1}$ with

$$\lambda2_{i+1} = \Delta\lambda_{i+1} + \lambda1_{i+1},$$  (23)

[revised manuscript text omitted]

---

## Author Response (AR2)

Authors' response to the second review of **Polarimetric radar reveals the spatial distribution of ice fabric at domes and divides in East Antarctica**

Manuscript Number: **tc-2020-370**

**Editor: Adam Booth**
**Referee #1: Thomas Jordan 24.10.2021**
**Referee #3: Emma C. Smith 24.11.2021**

Dear editor and reviewers,

thank you for the second round of insightful reviews. We have answered (AC) all of the reviewer's comments (RC) and implemented most of them. For brevity, we exclusively list non-editorial comments & answers below and mark the editorial remarks additionally in the track changes version. We thank all the reviewers and the editors for the detailed remarks throughout this review. Given that the last round of comments contained only minor points, this revision's core message remains unchanged.

On behalf of all co-authors,
M. Reza Ershadi
(University of Tübingen) 01.02.2022

**AC to RC1 (Tome Jordan):**

RC. *Anisotropic scattering parameter.* This has now been implemented by taking the positive square-route of the intensity ratio (eq. 19). I agree this is an improvement from before, as the scattering parameter is now linear w.r.t. the eigenvalue difference, and is therefore consistent with the matrix model scaling relationship. I am also happy to see that the suggested change did not have an adverse effect on the previous conclusions. *However, notation-wise I still think eq. (6) (where r is complex number) an eq. (19) (where r is a scalar) are inconsistent*. My recommendation would be to use |r| instead of r in eq. (19) (and all subsequent usage) and be clearer throughout the MS that it represents a magnitude.

**AC.** Thanks for bringing this up. We further clarified this in the revised MS (line 153). As mentioned in Appendix B of the MS (line 465), we follow Ackley and Keliher (1979) and Fujita (2006) and neglect the conductivity term using only the real part of the complex amplitude reflection coefficients ($Gamma\_X$ and $Gamma\_Y$). Therefore, equation 6 in the MS is also scalar (same as eq 19 in the MS). We hope this solves the apparent inconsistency between equations 6 and 19.
* * *
RC. *Significance of recovering the vertical eigenvalue for ice-flow modeling.* I think this could be made more explicit, as this is a key step forward for what the radar can provide to ice-sheet models (and it is considerably more useful than the `horizontal eigenvalue difference' _in constraining rheology). Ice-sheet modelers' will be interested in this result, as the strength of the vertical eigenvalue is a strong control on how soft ice is to shearing within the ice column - e.g. see: Azuma 1996 - https://www.cambridge.org/core/journals/annals-of-glaciology/article/an-anisotropic-flow-law-for-icesheet-ice-and-its implications/A9AC75ED14AD578C809CCF9752BF1D20

**AC.** We added this point and the mentioned reference to the revised MS abstract (line 6) and conclusion (line 438)**.**

**AC to RC3 (Emma Smith):**

Thank you for the editorial corrections and the very constructive comments. We implemented most of the suggested editorial changes, which you can find in the track change version. The parts with comments and questions are answered below.
* * *
RC. I would like to see a sentence here introducing the radar method briefly. As it is, it feels like it jumps into the technical part too quickly and i found it a bit tricky to understand what was going on until later into the abstract.

**AC.** We changed the radar polarimetry to co- and cross-polarized phase-sensitive radar data to clarify the radar method in revised MS (line 4).
* * *
**RC.** You mention this here, but don't go on to say why your method is comparatively advantageous, as you do with cores and borehole measurements. For completeness, it would make semse to mention the advantages of radar over surface sesimic methods in the next paragraph.

**AC.** We added this point to the revised MS (line 46).
* * *
**RC.** It seems as though you refer to Dome C/EDC and Kohnen/EDML interchangebly in this section and later on in thr mauscript. It is a little confusing, i would suggest you check for consistency throughout. There maybe be a reason to use a specific term in some places but it isnt completely clear to me.

**AC.** Excellent point. Thanks for mentioning it. In this paper, we have two radar sites at the location of the ice cores. EPICA Dome C, which we refer to as EDC and EPICA Dronning Maud Land, which we refer to as EDML. For Dome C, we also have another 19 radar sites. When we use Dome C, we mean the Dome C area and all the radar sites, but when we mention EDC, that means only the EPICA Dome C. For EDML, we do not have this because there is only one radar site. We went through the MS to make sure we correctly used these terms.
* * *
**RC.** Would be visually helpful if the site names were written at the top of each pannel - for quick reference.

**AC.** We decided not to add any more information to figure 1 since it already shows the sites' names with the coloured stars.
* * *
**RC.** A single ice crystal requires 60 times more stress to deform it in a non-basal direction, than along the basal plane (basal slip). This will not directly translate to a bulk ice fabric - i would suggest rephrasing this sentence to reflect this more accurately.

**AC.** This part has been rephrased in the revised MS (line 87).
* * *
**RC.** Is this beacause there is abrupt variability weith depth at EDML? If so, state here.

**AC.** Yes and we added this to the revised MS (line 236).
* * *
**RC.** State briefly why you compare to Dome Fuji here.

**AC.** Simply because it is another ice dome. As the contrasting flow regimes are elaborated on later in the MS, we left it here as is.
* * *
**RC.** Could you comment on ice flow environments where you might get a strong change in orientation with depth and how this limits the current method, in terms of environments where this can be applied?

**AC.** At the moment, it is not fully clear to us if a changing ice-fabric orientation should be interpreted as an ice dynamic signal of the current flow regime or if it indicates a temporal change of the flow regime at this location. Therefore it is difficult to single out specific cases where this method may not be applicable.
* * *
**RC.** Confusing sentence, i am not sure what you are referring to by "bulk of the fabric" - do you mean bulk ice fabric? Please clarify and rephrase.

**AC.** Yes. We meant bulk ice fabric, and it is rephrased in the revised MS (line 413).